# Integral Probability Metrics PAC-Bayes Bounds

**Ron Amit**
Technion - Israel Institute of Technology
The Viterbi Faculty of Electrical Engineering
`ronamit@campus.technion.ac.il`

**Baruch Epstein**
Technion - Israel Institute of Technology
The Viterbi Faculty of Electrical Engineering
`baruch.epstein@gmail.com`

**Shay Moran**
Technion - Israel Institute of Technology
Faculty of Mathematics
The Taub Faculty of Computer Science
Google Research, Israel
`smoran@technion.ac.il`

**Ron Meir**
Technion - Israel Institute of Technology
The Viterbi Faculty of Electrical Engineering
`rmeir@ee.technion.ac.il`

## Abstract

We present a PAC-Bayes-style generalization bound which enables the replacement of the KL-divergence with a variety of *Integral Probability Metrics* (IPM). We provide instances of this bound with the IPM being the *total variation metric* and the *Wasserstein distance*. A notable feature of the obtained bounds is that they naturally interpolate between classical uniform convergence bounds in the worst case (when the prior and posterior are far away from each other), and improved bounds in favorable cases (when the posterior and prior are close). This illustrates the possibility of reinforcing classical generalization bounds with algorithm- and data-dependent components, thus making them more suitable to analyze algorithms that use a large hypothesis space.

## 1 Introduction and Related Work

Classical statistical learning theory is based on a worst-case perspective which can be too pessimistic to model practical machine learning. In reality, data is rarely worst-case, and experiments demonstrate learning tasks that are solved with much less data than predicted by traditional theory. A primary manifestation of the traditional worst-case perspective is demonstrated by *uniform convergence* (UC); a genre of generalization bounds which form the backbone of the classical theory (Vapnik, 1999). These bounds guarantee that the generalization gap of *all* hypotheses in the output-space of the algorithm *simultaneously* vanish as the training-set size grows. The key algorithmic insight these bounds provide is summarized by the *Empirical Risk Minimization* principle (ERM), which asserts that it suffices to output *any* hypothesis in the class which minimizes the empirical risk.

Consequently, UC arguments provide non-trivial guarantees only if the hypothesis class used by the algorithm is *restricted* (e.g. has low Rademacher complexity or bounded VC dimension). In contrast, practical learning approaches such as deep learning algorithms use huge hypothesis classes whose VC dimensions rapidly increase with the size and depth of the underlying network. Hence, the rate guaranteed by UC arguments is often much slower than the rate observed in practice (Bachmann, Moosavi-Dezfooli, & Hofmann, 2021; Nagarajan & Kolter, 2019b; Neyshabur, Bhojanapalli, McAllester, & Srebro, 2017; C. Zhang, Bengio, Hardt, Recht, & Vinyals, 2017).

A further shortcoming of UC bounds, and the associated ERM principle, is that they are algorithm- and data-independent;[1] that is, they do not utilize beneficial properties of the data and/or the algorithm. For

---

[1]More precisely, UC bounds only depend on the hypothesis space.

36th Conference on Neural Information Processing Systems (NeurIPS 2022).

example, in practice, regularized algorithms often perform better than Empirical Risk Minimization, but this cannot be expressed by UC bounds and the ERM principle.

The PAC-Bayes (PB) framework is a prominent example of a theoretical framework that does not require the UC property. This framework was pioneered by Shawe-Taylor and Williamson (1997) and McAllester (1998) and developed in later papers, e.g. (Catoni, 2007; Lever, Laviolette, & Shawe-Taylor, 2013; Maurer, 2004; McAllester, 2003; Seeger, 2002); see Guedj (2019) and Alquier (2021) for extensive surveys. Begin, Germain, Laviolette, and Roy (2016) introduced a general strategy that produces PB bounds from change-of-measure inequalities leading to bounds based on the Rényi's $\alpha$-divergence, and Alquier and Guedj (2018); Ohnishi and Honorio (2021); Picard-Weibel and Guedj (2022) further extended PB bounds to other Csiszár's $f$-divergences.

PB theorems consider the generalization performance of stochastic predictors. These bounds are non-uniform[2] by nature, and are algorithm and data-dependent. They are usually based on a complexity term that depends on the Kullback-Leibler (KL) divergence between a data-dependent posterior distribution and a data-independent prior distribution.[3]

There are additional notable works on data and algorithm-dependent guarantees. The classical work of Bousquet and Elisseeff (2002) and Xu and Mannor (2012) studied generalization guarantees that depend on data and algorithm-dependent stability measures. A further line of recent papers tries to incorporate noise robustness/resilience. In Miyaguchi (2019), a PAC-Bayes transportation bound is used to measure the contribution of randomization to PB. This is done via optimal transport and Lipschitzness, based on the usual KL-PB bound. The work of Wei and Ma (2019) uses data-dependent Lipschitz smoothness to improve margin bounds, and Nagarajan and Kolter (2019a) passes from standard PB to a deterministic bound by assuming noise-resilience on the training data. This property translates to the test data, implying that good training smoothness leads to good test smoothness. Finally, Yang, Sun, and Roy (2019) measure data-dependent smoothness around each hypothesis (for each sample) and merge Catoni's bound (Catoni, 2007) with Rademacher theory, to obtain fast rates.

Recent work by Aminian, Bu, Wornell, and Rodrigues (2022); Lopez and Jog (2018); Rodríguez Gálvez, Bassi, Thobaben, and Skoglund (2021); Wang, Diaz, Santos Filho, and Calmon (2019); J. Zhang, Liu, and Tao (2021) and Neu and Lugosi (2022) proved information-theoretic bounds on the *expected* generalization gap using the Wasserstein and the total-variation (TV) distances. Our work is within the PB framework, and therefore enjoys the following advantages: *(i)* The bounds are "in high probability" over the sample rather than in expectation. *(ii)* PB bounds are sample dependent, i.e., bound the generalization gap for a specific sample-dependent posterior, while information-theoretic bounds are formulated as expectation over all sample sets, thereby providing a basis for empirical algorithms, e.g., Alquier (2021); Dziugaite and Roy (2017). *(iii)* The reference measure in PB can be any sample-independent distribution, while information-theoretic bounds consider a specific reference. Our work introduces uniform convergence assumptions, while the above-mentioned papers each used different assumptions. Recently, Chee and Loustau (2021) proposed PB bounds with the entropy regularized optimal transport distance for an online-learning setting with a finite class.

The optimal transport interpretation of the Wasserstein distance has been used recently in other contexts to derive generalization bounds. Chuang, Mroueh, Greenewald, Torralba, and Jegelka (2021) proposed a bound that uses a data-dependent complexity measure, evaluated via the Wasserstein distance of independently sampled subsets of the training data in the feature space. Hou, Kassraie, Kratsios, Rothfuss, and Krause (2022) used the principles of optimal transport to derive an instance-based bound based on the local Lipschitz regularity of the learned prediction function in the data space.

In the modern deep learning regime, measures of the hypothesis class complexity used in UC bounds, such as the VC dimension or Rademacher complexity, are enormous, making the bounds extremely loose for any reasonable number of samples, as opposed to PB bounds (Dziugaite & Roy, 2017; Jiang, Neyshabur, Mobahi, Krishnan, & Bengio, 2019). However, these complexity measures often have closed-form formulas for models such as neural networks, which show explicitly the effect of the model architecture (number of layer, activation functions etc.). This in contrast to PB bounds, in which the dependence on the hypothesis class is less explicit (but see Anthony and Bartlett

---

[2]I.e., PB bounds apply even in learning problems without uniform convergence (Definition 1).
[3]But see Rivasplata, Kuzborskij, Szepesvári, and Shawe-Taylor (2020) for data-dependent priors.

(1999); Neyshabur, Tomioka, and Srebro (2015) for exceptions for neural networks). Therefore, we believe that extension of UC bounds to incorporate data- and algorithm-dependence can facilitate the design of better performing architectures. A further advantage of PB bounds is their non-uniformity (the generalization gap bound depends on the learning output), hence we can use the bound as a minimization objective for a structural minimization algorithm, where the complexity term acts as a regularizer. In cases where the hypothesis class is very large, but we have some prior knowledge on which hypotheses are more likely to have low population loss (e.g. prefer simpler hypothesis as suggested by Occam's Razor), then in PB one can inject this knowledge as the prior distribution, effectively lowering the generalization bound.

Can the rich theory of UC bounds be extended to help explain generalization with modern large scale models? Can this theory be used to prove data and algorithm dependent guarantees? In this paper, we take a step in the direction of answering these questions positively. To achieve this goal, we show a new technique to incorporate UC bounds within the PAC-Bayes framework. We prove new PB bounds with Integral Probability Metric (IPM) (Müller, 1997) to measure distances between distributions, rather than the standard KL or $f$-divergences used so far. Specifically, we focus on utilizing two specific IPMs: the total variation and Wasserstein metrics This IPM framework allows greater flexibility, as it does not require the support of the posterior to be a subset of the prior's support (absolute continuity) as in standard KL-PB bounds, and it applies to deterministic as well as stochastic prior and posterior distributions. In fact, the IPM-based PB bounds we introduce match, at worst, the rate of the UC bound used. Recently, Livni and Moran (2020) showed that the classical KL-PB theorem cannot imply meaningful distribution-free generalization bounds for 1-dimensional linear classification. In contrast, our derived IPM-PB bounds do imply such bounds, because linear classifiers satisfy uniform convergence.

We note that the work of Audibert and Bousquet (2003, 2007) showed a different approach to utilizing the UC assumption to derive PAC-Bayes bounds. Their work assumed a UC property to utilize the generic chaining technique, resulting in more refined, variance-sensitive, bounds. In contrast to our work, their bound is not fully empirical, and the assumed UC bound is not used by the resulting bound.

The Total Variation PAC-Bayes (TVPB) bound (Thm. 6) applies in any setting where uniform convergence (UC) (Def. 1) holds, an assumption that is satisfied by many natural learning problems. For example, in binary problems, UC holds with a rate of $O(\sqrt{\mathtt{VC}(\mathcal{H})/m})$, where $\mathtt{VC}(\mathcal{H})$ is the VC dimension (Vapnik, 1999). As observed in practice, for large models of deep neural networks with very large VC dimensions the learning rate on natural datasets is often much faster than predicted UC bounds. To explain this gap, we must turn to data and algorithm-dependent bounds. The TVPB improves the gap bound to be $O\left(\sqrt{\mathtt{VC}(\mathcal{H})D_{\mathrm{TV}}(Q,P)/m}\right)$, effectively multiplying the VC dimension by the total variation distance of the posterior from the prior. Intuitively, simpler posteriors (closer to the prior assumed before observing the data) lead to a better generalization gap. Compared to the vanilla KL-PB, the TV distance can be small even in cases where the KL distance can be very large, and in any case, the bound only improves over the original UC bound. In addition, the TV-PB bounds incorporate properties of $\mathcal{H}$ via the VC dim. We also explore settings where the generalization gap function exhibits a certain smoothness property, and show a PB bound with the Wasserstein metric (Thm. 11), We analyze this smoothness property and show an explicit Wasserstein-based bound in the finite hypothesis class setting and in a linear regression setting. In the latter setting, we show that a standard UC bound can be improved by a factor of $O(\sqrt{W_1(Q,P)})$, where $W_1(Q,P)$ is the 1st order Wasserstein distance between posterior and prior over the unit-sphere. We conduct a numerical simulation to demonstrate the improvement of the Wasserstein PB bound over the UC bound, and, in cases of narrow prior distributions, over the KL-PB bound. The experiment also investigates the case of non-randomized predictors by setting the prior and posterior as Dirac delta measures, which $f$-divergence based PB bounds are unable to use.

Figure 1 illustrates graphically the organization of the claims in the paper.

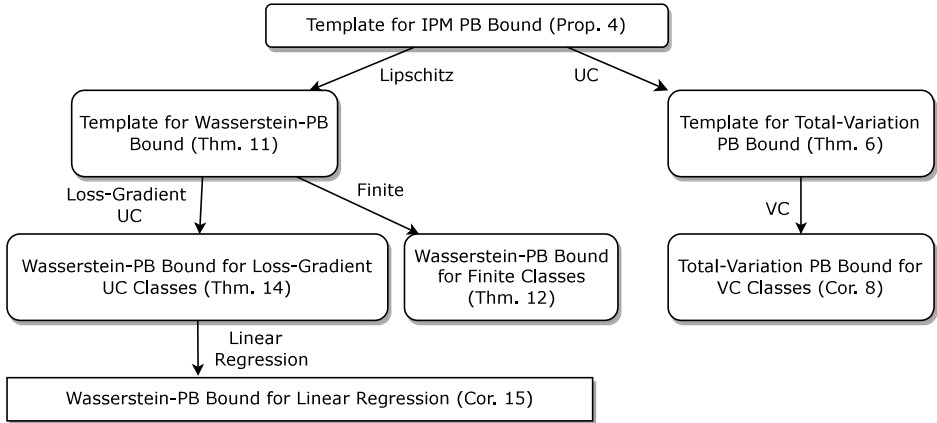

Figure 1: Claims tree diagram. $A \rightarrow B$ means that claim $B$ is a special case of claim $A$, under additional assumptions on the learning problem. For full description of the assumptions, see the corresponding claims.

## 2 Preliminaries

### 2.1 The Learning Problem

We begin with a short description a standard supervised learning task. Consider a domain $\mathcal{Z}$, [4] a distribution $\mathcal{D}$ over $\mathcal{Z}$, a hypothesis set $\mathcal{H}$, and finally, a loss function $\ell : \mathcal{H} \times \mathcal{Z} \rightarrow [0, 1]$. The tuple $(\mathcal{D}, \mathcal{H}, \ell)$ defines a learning problem: The learning algorithm receives as input a training set $S = \{z_i\}_{i=1}^{m} \in \mathcal{Z}^m$ sampled i.i.d from $\mathcal{D}$ and selects an hypothesis $h \in \mathcal{H}$. The performance of $h$ is measured by the *expected risk*, $L_{\mathcal{D}}(h) \stackrel{\text{def}}{=} \mathbb{E}_{z \sim \mathcal{D}} \ell(h, z)$. While the expected risk is unavailable to algorithm, the *empirical risk*, $\hat{L}_S(h) \stackrel{\text{def}}{=} \frac{1}{m} \sum_{i=1}^{m} \ell(h, z_i)$, can be evaluated using the training data. The generalization gap is defined by $\Delta_S(h) \stackrel{\text{def}}{=} L_{\mathcal{D}}(h) - \hat{L}_S(h)$.

Several of our results will assume that the learning problem $(\mathcal{D}, \mathcal{H}, \ell)$ satisfies the *uniform convergence property*; i.e. the existence of a uniform upper bound on the generalization gap which applies simultaneously to all hypotheses in $\mathcal{H}$.

**Definition 1** (Uniform convergence, ([Vapnik & Chervonenkis, 2015](#))). *The learning problem $(\mathcal{D}, \mathcal{H}, \ell)$ satisfies the uniform convergence property, if there exists a bound function $u(m, \delta') > 0$ s.t. for any $m \in \mathbb{N}^+$, $\delta \in (0, 1)$ we have*

$$\mathbb{P}\{|\Delta_S(h)| \leq u(m, \delta), \forall h \in \mathcal{H}\} \geq 1 - \delta \quad and \quad u(m, \delta) \underset{m \to \infty}{\rightarrow} 0. \quad (1)$$

UC type bounds are a major part of the foundations of theoretical machine learning. Unfortunately, they suffers from a few drawbacks. First, currently known bounds tend to be extremely loose in many cases, most notably for deep networks. Second, the setup does not provide a natural way to encode prior knowledge into the bounds, particularly when dealing with deep networks - the hypothesis set is usually rich enough to express all relevant functions, and the training algorithms that might utilize some prior knowledge are not themselves a part of UC based bounds. Finally, UC bounds are usually not data-dependent, a property which is critical to explain the generalization of DNNs on real-world data ([Nagarajan & Kolter, 2019b](#); [C. Zhang et al., 2017](#)).

### 2.2 PAC-Bayes Bounds

Let $\mathcal{M}(\mathcal{H})$ denote the set of all probability measures over $\mathcal{H}$. For any probability measure $Q \in \mathcal{M}(\mathcal{H})$, we define the expected loss, empirical loss and generalization gap by

$$L_{\mathcal{D}}(Q) \stackrel{\text{def}}{=} \underset{h \sim Q}{\mathbb{E}} L_{\mathcal{D}}(h) \quad ; \quad \hat{L}_S(Q) \stackrel{\text{def}}{=} \underset{h \sim Q}{\mathbb{E}} \hat{L}_S(h) \quad ; \quad \Delta_S(Q) \stackrel{\text{def}}{=} L_{\mathcal{D}}(Q) - \hat{L}_S(Q). \quad (2)$$

---

[4]This formulation allows for greater generality than the standard $\mathcal{Z} = \mathcal{X} \times \mathcal{Y}$ and mis-classification loss setting. In particular, it can describe a number of unsupervised learning problems ([Seldin and Tishby (2010)](#)).

PAC-Bayes theory bounds the expected loss, simultaneously for all "posterior" (sample-dependent) probability measures $Q \in \mathcal{M}(\mathcal{H})$, with high probability over the samples $S \sim \mathcal{D}^m$, given any "prior" (sample-independent) probability measure $P \in \mathcal{M}(\mathcal{H})$. A key feature of most PAC-Bayes bounds is their dependence on the KL divergence between the two distributions $P, Q$, $\text{KL}(Q \parallel P) \stackrel{\text{def}}{=} \int_{\mathcal{H}} \ln(\frac{\mathrm{d}Q}{\mathrm{d}P})\mathrm{d}Q$, where $\frac{\mathrm{d}Q}{\mathrm{d}P}$ is the Radon–Nikodym derivative of $Q$ w.r.t. $P$. While KL is a natural measure of divergence between probability distributions, it restricts the applicability of the resulting bounds to cases where the support of $Q$ is contained in the support of $P$. The following bound was introduced by McAllester (1998).

**Proposition 2** (Classical KL-PB Bound). *For any prior $P \in \mathcal{M}(\mathcal{H})$ and $\delta \in (0, 1)$, with probability at least $1 - \delta$ over the samples $S \sim \mathcal{D}^m$, for all $Q \in \mathcal{M}(\mathcal{H})$, we have*

$$\Delta_S(Q) \leq \sqrt{\frac{\text{KL}(Q \parallel P) + \ln(m/\delta)}{2(m-1)}}. \tag{3}$$

## 3 A Template for IPM PAC-Bayes Bounds

### 3.1 General IPM-PB Bound

**Definition 3** (Integral Probability Metric). *(Müller, 1997; Sriperumbudur, Fukumizu, Gretton, Schölkopf, & Lanckriet, 2009, 2012) The Integral Probability Metric (IPM) between two probability measures $P$ and $Q$ over $\mathcal{H}$ is defined as*

$$\gamma_{\mathcal{F}}(Q, P) \stackrel{\text{def}}{=} \sup_{f \in \mathcal{F}} \left| \int_{\mathcal{H}} f \mathrm{d}P - \int_{\mathcal{H}} f \mathrm{d}Q \right| = \sup_{f \in \mathcal{F}} \left| \underset{h \sim P}{\mathbb{E}}[f(h)] - \underset{h \sim Q}{\mathbb{E}}[f(h)] \right|, \tag{4}$$

*where $\mathcal{F}$ is a set of real-valued bounded functions $\mathcal{H} \to \mathbb{R}$.*

By definition, IPM distance measures are symmetric and non-negative. Note that the KL-divergence is not a special case of IPM, rather it belongs to the family of $f$-divergences, that intersect with IPM only at the Total-Variation (Sriperumbudur et al., 2009, 2012).

The following proposition assumes that for any fixed sample $S' \in \mathcal{Z}^m$, the function $f_{S'}(h) \stackrel{\text{def}}{=} 2(m-1)\Delta_{S'}^2(h)$ is a member of a family of functions that depend on the sample, denoted $\mathcal{F}_{S'}$. Thus, the IPM-PB bound allows us to 'convert' some knowledge we have about the properties of the generalization gap function $\Delta_S(h)$ to a generalization bound.

Since we do not specify yet the collection of function families $\{\mathcal{F}_S\}_{S \in \mathcal{Z}^m}$, the bound does not convey an explicit rate, and it should rather be seen as a **template**. In the next sections, we will derive explicit bounds with specific IPMs divergences. Namely, we will derive a total-variation distance based bound by selecting a collection of bounded function sets, and a Wasserstein distance based bound, by selecting a collection of Lipschitz function sets.

**Proposition 4** (Template for IPM PB Bound). *For any fixed dataset $S' \in \mathcal{Z}^m, m \in \mathbb{N}^+$, let $\mathcal{F}_{S'}$ be a family of bounded and measurable functions $\mathcal{H} \to \mathbb{R}$. Assume that for any number of samples, $m$, and sample $S' \in \mathcal{Z}^m$, the function $2(m-1)\Delta_{S'}^2(\cdot)$ is in $\mathcal{F}_{S'}$. Then for any prior $P \in \mathcal{M}(\mathcal{H})$ and $\delta \in (0, 1)$, with probability at least $1 - \delta$ over the samples $S \sim \mathcal{D}^m$, for all $Q \in \mathcal{M}(\mathcal{H})$, we have*

$$\Delta_S(Q) \leq \sqrt{\frac{\gamma_{\mathcal{F}_S}(Q, P) + \ln(m/\delta)}{2(m-1)}}. \tag{5}$$

The proof is in Appendix A.1. The main idea is to use the IPM definition and the assumption as a change-of-measure inequality, instead of the variational formula by Donsker and Varadhan (1975), which is used in the classical KL-PB bound proof. The rest of the proof is similar to the classical derivation (McAllester, 2003; Shalev-Shwartz & Ben-David, 2014).

Note that, similarly to the classical KL-PB bound, Proposition 4 does not require the UC property to hold. However, in the next sections we will see that assuming an existence of a UC bound $u(m, \delta)$, and selecting a particular collection of function families $\{\mathcal{F}_S\}_{S \in \mathcal{Z}^m}$, will result in explicit bounds that can improve upon the worst-case nature of the original $u(m, \delta)$ bound.

## 3.2 Template for Seeger Type IPM PAC-Bayes Bound

The work of Seeger (2002) and Maurer (2004) presented a different form of the PAC-Bayes theorem with fast $O(1/m)$ rate as the dominant term, if the empirical risk is low.

We denote by $\mathtt{kl}(p \parallel q)$ the KL divergence between two Bernoulli distributions $\mathcal{B}(p)$ and $\mathcal{B}(q)$, $p, q \in [0, 1]$, that is,

$$\mathtt{kl}(p \parallel q) \overset{\text{def}}{=} \begin{cases} p \ln\left(\frac{p}{q}\right) + (1-p)\ln\left(\frac{1-p}{1-q}\right), & \text{if } q \notin \{0, 1\} \\ 0, & \text{else if } p = q \\ \text{undefined}, & \text{else.} \end{cases} \tag{6}$$

For any $h \in \mathcal{H}$, define the relative entropy of the empirical risk with respect to the expected one as $\Delta_S^{\mathtt{kl}}(h) \overset{\text{def}}{=} \mathtt{kl}(\hat{L}_S(h) \parallel L_D(h))$. Similarly, and overloading notations, we define for any distribution $Q \in \mathcal{M}(\mathcal{H})$, $\Delta_S^{\mathtt{kl}}(Q) \overset{\text{def}}{=} \mathtt{kl}(\hat{L}_S(Q) \parallel L_D(Q))$.

By replacing the KL-based change-of-measure inequality step in the proof of (Maurer, 2004, Thm. 5) with the IPM property (Def. 3) we get a similar bound for IPM measures (see a detailed proof in Appendix A.2).

**Proposition 5** (Template for Seeger Type IPM PAC-Bayes Bound). *Assume $f_S(h) \overset{def}{=} m \cdot \Delta_S^{\mathtt{kl}}(h) \in \mathcal{F}_S$. Then for any prior $P \in \mathcal{M}(\mathcal{H})$ and $\delta \in (0, 1)$, with probability at least $1 - \delta$ over the samples $S \sim \mathcal{D}^m$, for all $Q \in \mathcal{M}(\mathcal{H})$, we have*

$$\Delta_S^{\mathtt{kl}}(Q) \leq \frac{\gamma_{\mathcal{F}_S}(Q, P) + \ln(2\sqrt{m}/\delta)}{m}. \tag{7}$$

*By applying the Refined Pinsker's relaxation (McAllester (2003), Eq. 6) we can immediately derive the following, looser, but easier to interpret bound*

$$\Delta_S(Q) \leq \sqrt{2\hat{L}_S(Q)\frac{\gamma_{\mathcal{F}_S}(Q, P) + \ln(2\sqrt{m}/\delta)}{m}} + 2\frac{\gamma_{\mathcal{F}_S}(Q, P) + \ln(2\sqrt{m}/\delta)}{m}. \tag{8}$$

When $\hat{L}_S(Q)$ is small (as is typical with modern deep networks), the final term determines the convergence rate. We defer the investigation of PB bounds derived from Prop. 5 to Appendix B. In the following sections we focus on investigating the implication of the Template IPM PB Bound of Prop. 4.

# 4 Total-Variation PAC-Bayes Bounds

In this section, we investigate a PB bound with the total-variation (TV) distance, $D_{\text{TV}}(Q, P) \overset{\text{def}}{=} \sup_{A \in \Sigma_{\mathcal{H}}} |P(A) - Q(A)|$, where $\Sigma_{\mathcal{H}}$ is the standard Borel sigma-algebra associated with $\mathcal{H}$. The TV distance can be described as an IPM with the family of functions

$$\mathcal{F}_M^\infty \overset{\text{def}}{=} \{f : \mathcal{H} \to [0, \infty), \|f\|_\infty \leq M\}, \tag{9}$$

for any $M \geq 0$. To see this, note that

$$\gamma_{\mathcal{F}_M^\infty}(Q, P) = \sup_{f \in \mathcal{F}_M^\infty} \left| \int_{\mathcal{H}} f \mathrm{d}P - \int_{\mathcal{H}} f \mathrm{d}Q \right| \overset{(i)}{=} M \cdot \sup_{A \in \Sigma_{\mathcal{H}}} |P(A) - Q(A)| = MD_{\text{TV}}(Q, P), \tag{10}$$

where equality *(i)* holds since in the supremum it suffices to take the class of indicator functions $\{M \cdot \mathbb{1}_A(h), A \in \Sigma_{\mathcal{H}}\}$, since the functions in $\mathcal{F}_M^\infty$ are bounded in $[0, M]$.

**Theorem 6** (Template for Total-Variation PB Bound). *Assume that there exists some uniform convergence bound $u(m, \delta')$ (Definition 1), then, for any prior $P \in \mathcal{M}(\mathcal{H})$ and $\delta \in (0, 1)$, with probability of at least $1 - \delta$ over samples $S \sim \mathcal{D}^m$, for all $Q \in \mathcal{M}(\mathcal{H})$, we have*

$$\Delta_S(Q) \leq \sqrt{u^2(m, \delta/2)D_{\text{TV}}(Q, P) + \frac{\ln(2m/\delta)}{2(m-1)}}. \tag{11}$$

The proof (Appendix A.3) follows directly from the general IPM PB bound (Prop. 4) and the uniform convergence assumption, using a union bound argument. This bound can be seen as a template to be used to derive explicit PB bounds, by plugging in existing UC bounds. Note that while we require the existence of UC bound, the resulting bound is nonuniform (since it depends on the data-dependent posterior).

Compared to the original UC bound, $u(m, \delta)$, the bound in (11) is roughly multiplied by a factor of $\sqrt{D_{\mathrm{TV}}(Q, P)} \in [0, 1]$, ensuring tighter guarantees, especially if the posterior is close to the prior.

For example, consider a binary classification case, with the zero-one loss function and $\mathtt{VC}(\mathcal{H})$ class $\mathcal{H}$. The well-known UC theorem states that the generalization gap converges uniformly at a rate $O\left(\sqrt{\mathtt{VC}(\mathcal{H})/m}\right)$.

**Proposition 7** (VC Bound, Boucheron, Bousquet, and Lugosi (2005)). *There exists some universal constant [5] $c > 0$ s.t. for any $\delta \in (0, 1)$ we have*

$$\mathbb{P}\left\{\Delta_S(h) \leq c\sqrt{\frac{\mathtt{VC}(\mathcal{H}) + \ln(1/\delta)}{m}}, \forall h \in \mathcal{H}\right\} \geq 1 - \delta. \tag{12}$$

Using Thm. 6, we derive the following algorithm and data-dependent bound.

**Corollary 8** (Total-Variation PB Bound for VC Classes). *Consider a binary classification problem, with the zero-one loss, and hypothesis class $\mathcal{H}$, with finite VC dimension, $\mathtt{VC}(\mathcal{H})$. There exists some universal constant $c > 0$ s.t. for any prior $P \in \mathcal{M}(\mathcal{H})$ and $\delta \in (0, 1)$, with probability of at least $1 - \delta$ over samples $S \sim \mathcal{D}^m$, for all $Q \in \mathcal{M}(\mathcal{H})$, we have*

$$\Delta_S(Q) \leq \sqrt{c\frac{\mathtt{VC}(\mathcal{H}) + \ln(1/\delta)}{m}D_{\mathrm{TV}}(Q, P) + \frac{\ln(m/\delta)}{2(m-1)}}. \tag{13}$$

Compared to the UC bound of Prop. 7, Cor. 8 multiplies the dominant term of the bound by a nonuniform (data and algorithm-dependent) factor of $\sqrt{D_{\mathrm{TV}}(Q, P)}$, which is guaranteed to tighten the bound.

Note that the total-variation based bound of Aminian et al. (2022) and Rodríguez Gálvez et al. (2021) assume Lipschitz loss function, while our TV bound allows non continuous loss functions such as the zero-one loss. The TV based bound of Wang et al. (2019) (Thm. 1) is not directly comparable, since the empirical risk term is multiplied by a factor that goes to infinity for TV distance that goes to 1.

## 5 Wasserstein PAC-Bayes Bounds

### 5.1 Template for Wasserstein-PB Bound

In this section, we provide a PAC-Bayes generalization bound with the Wasserstein metric between posterior and prior and a certain smoothness parameter of the generalization gap function. We explore learning settings for which $\mathcal{H}$ can be paired with a distance metric $\rho : \mathcal{H} \times \mathcal{H} \to \mathbb{R}_{\geq 0}$ s.t. $(\mathcal{H}, \rho)$ is a Polish metric space (complete, separable metric space).[6] Given the distance metric $\rho$, we can define the Wasserstein distance between any two probability measures on $\mathcal{H}$.

**Definition 9** (Wasserstein Distance). *For any two probability measures $P, Q$ on $\mathcal{H}$ with finite first moment, the $1^{st}$ order Wasserstein distance is*

$$W_1(Q, P) \stackrel{def}{=} \inf_{\gamma \in \Gamma(Q, P)} \int_{\mathcal{H} \times \mathcal{H}} \rho(h, h')\mathrm{d}\gamma(h, h'), \tag{14}$$

*where $\Gamma(Q, P)$ denotes the set of all couplings of $Q$ and $P$, that is, the set of all joint measure on $\mathcal{H} \times \mathcal{H}$ whose marginals are $Q$ and $P$.*

---

[5]The bound of Cor. 7 originates from Talagrand (1994). As far as we know, there is no explicit value of the universal constant in the literature. Obtaining the constant involves careful computations of covering numbers and using the chaining method (e.g., based on Thm. 1.16 and 1.17 in Lugosi (2002)). Since our focus was not on the numerical evaluation of the bounds, we did not include this in our work. We note that there are other VC-type bounds with explicit constants, but with an extra $\log(m)$ factor (e.g., Vapnik (1999), Sect 3.4).

[6]See Villani (2006) Ch. 1, for a discussion of this assumption.

The following proposition gives a dual representation for the first-order Wasserstein distance.

**Proposition 10** (Kantorovich-Rubinstein Duality ([Villani, 2006])). *For any $0 \leq K$, and any two probability measures $P, Q \in \mathcal{M}(\mathcal{H})$,*

$$K \cdot W_1(Q, P) = \sup_{f \in \mathcal{F}_K^{Lip}} \left| \mathbb{E}_{h \sim P} [f(h)] - \mathbb{E}_{h \sim Q} [f(h)] \right|, \tag{15}$$

*where $\mathcal{F}_K^{Lip}$ is the set of $K$-Lipschitz functions w.r.t. $\rho(h, h')$, i.e. functions that satisfy*

$$\sup_{(h,h') \in \mathcal{H}^2} \frac{|f(h) - f(h')|}{\rho(h, h')} \leq K. \tag{16}$$

We can write the Kantorovich-Rubinstein duality (15) using IPM formulation (Def. 3),

$$K \cdot W_1(Q, P) = \gamma_{\mathcal{F}_K^{\text{Lip}}}(P, Q). \tag{17}$$

Using this duality we will prove the following bound.

**Theorem 11** (Template for Wasserstein-PB Bound). *Assume that for any $\delta' \in (0, 1]$, w.p. at least $1 - \delta'$ over the sampling $S \sim \mathcal{D}^m$, the squared generalization gap function, $\Delta_S^2(\cdot)$, is $K$-Lipschitz w.r.t. the metric $\rho$ with some $K = K(m, \delta')$. Then, for any prior $P \in \mathcal{M}(\mathcal{H})$ and $\delta \in (0, 1)$, with probability of at least $1 - \delta$ over samples $S \sim \mathcal{D}^m$, for all $Q \in \mathcal{M}(\mathcal{H})$, we have*

$$\Delta_S(Q) \leq \sqrt{K(m, \delta/2) W_1(Q, P) + \frac{\ln(2m/\delta)}{2(m-1)}}. \tag{18}$$

The proof (Appendix A.4) follows directly from Proposition 4 and the assumption, via the union bound. Theorem 11 can be seen as a template to be used for deriving Wasserstein-PB bounds in various learning settings where the $\Delta_S^2(\cdot)$ is $K$-Lipschitz with high probability (over samples), where the rate of $K = K(m, \delta/2)$ should be $O(1/m)$ to ensure a factor $O(1/\sqrt{m})$ multiplying the divergence between posterior and prior, as in the KL-PB bound.

Such a result can be challenging to prove since it requires uniform convergence of the slope between any two hypotheses in $\mathcal{H}$. Next, we will show specific learning settings where this property holds and the resulting generalization bounds.

We note that recent work Neu and Lugosi (2022) also establishes a Wasserstein-based information-theoretic generalization bound. This work assumed infinitely smooth loss functions and established bounds on the expected generalization gap, rather than high-probability bounds. In fact, Neu and Lugosi (2022) noted that obtaining such bounds is an open problem. Observe, though, that our bound depends on the Lipschitz constant $K = K(m, \delta)$ which needs to assessed; see sections 5.2 and 5.3 for specific examples. The general problem remains open.

At a more pragmatic level, we note that learning algorithms derived from minimizing Wasserstein based PB bounds have an added benefit of more stable optimization compared to KL based approaches, due to lower gradient variance, as noted in Arjovsky, Chintala, and Bottou (2017), and this distance measure can be approximated efficiently from finite samples (Cuturi, 2013; Weed & Bach, 2017)

## 5.2 Wasserstein-PB Bound for Finite Classes

We first investigate the simple case of a finite hypothesis class with a loss function $\ell$ which is $G$-Lipschitz w.r.t. the metric $\rho$. Note that for finite classes, UC always holds. We derive the following bound from Thm. 11.

**Theorem 12** (Wasserstein-PB Bound for Finite Classes). *Let $\mathcal{H}$ be a finite hypothesis class. Assume that for any fixed $z \in \mathcal{Z}$, $\ell(h, z)$ is a $G$-Lipschitz function in $h$ w.r.t the metric $\rho$. Then for any prior $P \in \mathcal{M}(\mathcal{H})$ and $\delta \in (0, 1)$, with probability of at least $1 - \delta$ over samples $S \sim \mathcal{D}^m$, for all $Q \in \mathcal{M}(\mathcal{H})$, we have*

$$\Delta_S(Q) \leq \sqrt{\frac{8G \log(4|\mathcal{H}|/\delta)}{m} W_1(Q, P) + \frac{\ln(2m/\delta)}{2(m-1)}}. \tag{19}$$

The proof (Appendix A.5) makes use of standard union bound arguments, Hoeffding's concentration inequality and the template Wasserstein-PB bound (Thm. 11). Notice that compared to the standard UC bound for finite classes, Thm. 12 multiplies the bound by a nonuniform factor of $\sqrt{GW_1(Q, P)}$.

### 5.3 Wasserstein-PB Bound for Loss-Gradient UC Classes

In this section we show a Wasserstein-PB bound for learning problems $(\mathcal{D}, \mathcal{H}, \ell)$, with $\mathcal{H} \subset \mathbb{R}^d$, for some dimension $d \in \mathbb{N}^+$, that satisfy the standard UC property, and, additionally, satisfy UC property for the loss gradient, as defined below.

**Definition 13** (Loss-Gradient UC Property). *A learning problem $(\mathcal{D}, \mathcal{H}, \ell)$, with $\mathcal{H} \subset \mathbb{R}^d$, is said to satisfy the **loss-gradient UC property**, if: (i) the loss function $\ell(h, z)$ is differentiable w.r.t. h on $Int(\mathcal{H}) \times \mathcal{Z}$ and continuous w.r.t. h on $\mathcal{H} \times \mathcal{Z}$. (ii) The problem satisfies the uniform convergence property (Def. 1). (iii) The empirical average of the loss gradient converges uniformly in $L_2$ norm sense to its mean. I.e., there exists a bound function $u^{grad}(m, \delta) > 0$, s.t. for any $\delta \in (0, 1)$ we have $u^{grad}(m, \delta) \underset{m \to \infty}{\to} 0$ and for any $m \in \mathbb{N}^+$,*

$$\mathbb{P}\left(\left\|\mathbb{E}_{z \sim \mathcal{D}} \nabla_h \ell(h, z) - \frac{1}{m} \sum_{i=1}^{m} \nabla_h \ell(h, z_i)\right\|_2 \leq u^{grad}(m, \delta), \forall h \in Int(\mathcal{H})\right) \geq 1 - \delta, \quad (20)$$

*where $\nabla_h \ell(h, z)$ denotes the gradient of $\nabla_h \ell(h, z)$ w.r.t. h, for a fixed $z \in \mathcal{Z}$, and $Int(\mathcal{H})$ is the interior of $\mathcal{H}$. We call $u^{grad}$, the UC bound of the loss gradient.*

**Theorem 14** (Wasserstein-PB Bound for Loss-Gradient UC Classes). *Let $(\mathcal{H}, \rho)$ be a metric space such that $\mathcal{H} \subset \mathbb{R}^d$ is a closed and convex set, and $\rho$ is the $L_2$ distance. Assume the learning problem satisfies the loss-gradient UC property (Def. 13), with UC bound $u$, and a UC bound of the loss gradient, $u^{grad}$. Then for any prior $P \in \mathcal{M}(\mathcal{H})$ and $\delta \in (0, 1)$, with probability of at least $1 - \delta$ over samples $S \sim \mathcal{D}^m$, for all $Q \in \mathcal{M}(\mathcal{H})$, we have*

$$\Delta_S(Q) \leq \sqrt{2 \cdot u(m, \delta/4) \cdot u^{grad}(m, \delta/4) \cdot W_1(Q, P) + \frac{\ln(2m/\delta)}{2(m-1)}}. \quad (21)$$

The proof (Appendix A.6) is derived from the assumptions and the template Wasserstein-PB bound (Thm. 11). In learning problems that satisfy the loss-gradient UC property, we often have $u(m, \delta/4), u^{grad}(m, \delta/4) \in O(1/\sqrt{m})$, and then the resulting bound is $O(1/\sqrt{m})$. We provide full analysis that shows such a rate for the following linear regression example.

### 5.4 Linear Regression Example

Based on the Wasserstein-PB Bound for Loss-Gradient UC Classes (Thm. 14), we derive the following corollary.

**Corollary 15** (Wasserstein-PB Bound for Linear Regression). *Consider a data distribution of pairs $z = (x, y)$, where $x$ is sampled from an unknown distribution supported on a $d$-dimensional ball of radius $r > 0$, $\mathbb{B}_r^d \overset{def}{=} \{x \in \mathbb{R}^d : \|x\|_2 \leq r\}$, and the target, $y = f(x)$, is set by an unknown, possibly random, target function $f : \mathbb{B}_r^d \to [-1, 1]$. The hypothesis space $\mathcal{H}$ is $\mathbb{B}_{1/r}^d$, and the loss function is $\ell(x, y, h) = \frac{1}{4}(h^\top x - y)^2$. Then, for any prior $P \in \mathcal{M}(\mathcal{H})$ and $\delta \in (0, 1)$, with probability at least $1 - \delta$ over samples $S \sim \mathcal{D}^m$, for all $Q \in \mathcal{M}(\mathcal{H})$,*

$$\Delta_S(Q) \leq \sqrt{2u(m, \delta/4) \cdot u^{grad}(m, \delta/4) \cdot W_1(Q, P) + \frac{\ln(2m/\delta)}{2(m-1)}}, \quad (22)$$

*where $W_1(Q, P)$ denotes the $1^{st}$ order Wasserstein distance with the $L_2$ metric,*

$$u(m, \delta) \in O\left(\sqrt{\frac{d(1 + \ln(1/\delta))}{m}}\right), \text{ and } u^{grad}(m, \delta) \in O\left(r\sqrt{\frac{d(1 + \ln(1/\delta))}{m}}\right). \quad (23)$$

The full expression of the bound appears in the theorem's proof (Appendix A.7). Ignoring logarithmic factors we obtained a UC bound of $\tilde{O}\left(\sqrt{\frac{d}{m}}\right)$, and a Wasserstein-PB bound of $\tilde{O}\left(\sqrt{rW_1(Q, P)\frac{d}{m}}\right)$. Note that from Thm. 6 we can also deduce a TV-PB bound of order $\tilde{O}\left(\sqrt{D_{\text{TV}}(Q, P)\frac{d}{m}}\right)$. In

comparison, the standard KL-PB bound is of order $\tilde{O}\left(\sqrt{\frac{\text{KL}(Q \parallel P)}{m}}\right)$. The TV-PB bound is, at worst, roughly the same as the UC bound, since $D_{\text{TV}}(Q, P) \leq 1$. Note that since $Q$ and $P$ are distributions over a sphere of radius $1/r$, then $rW_1(Q, P) \leq 2$. Hence, the Wasserstein-PB is also, at worst, roughly the same as the UC bound. However, the KL-PB bound can be either tighter or looser, depending on $P$ and $Q$. In cases where the mass of the posterior $Q$ is concentrated in a region of the hypothesis set where the prior $P$ is arbitrarily small, then the KL divergence can be arbitrarily large, making the KL-PB bound extremely loose compared to the UC, TV-PB, and Wasserstein-PB bounds. The numerical experiment described in Appendix C demonstrates this by investigating different prior distributions with different widths. In particular, for posteriors and priors that are Dirac delta distributions (i.e., deterministic predictors), we show that the Wasserstein-PB considerably improves over the UC bound, while the KL-PB bound is undefined. We therefore demonstrated non-vacuous guarantees for deterministic models within the PB framework without requiring additional derandomization steps.

We can compare the bound of Thm. 14 to the bound of Corollary 8, in Neu and Lugosi (2022), which is also dependent on the Wasserstein distance between a data-dependent output (posterior) and a base measure (prior). The bound of Thm. 14 is different in the sense that *(i)* It holds with high probability instead of in expectation. *(ii)* Instead of assuming infinitely-smooth loss function with $\beta$-bounded directional derivatives, Thm. 14 assumes a loss-gradient UC class. *(iii)* The bound scales as $\tilde{O}\left(\sqrt{W_1 \cdot u \cdot u^{\text{grad}}}\right)$ instead of $\tilde{O}\left(\sqrt{W_2 \cdot \frac{d\beta}{m}}\right)$.

In our linear regression example, Cor. 15 scales as $\tilde{O}\left(\sqrt{W_1 \cdot u \cdot u^{\text{grad}}}\right) = \tilde{O}\left(\sqrt{r\sqrt{\frac{d}{m}} r \sqrt{\frac{d}{m}}}\right) = \tilde{O}\left(r\sqrt{\frac{d}{m}}\right)$. The loss is infinitely-smooth with $\beta \in O(1 + r + r^2)$, and therefore Cor. 8 of Neu and Lugosi (2022) scales as $\tilde{O}\left(\sqrt{W_2 \cdot \frac{d\beta}{m}}\right) = \tilde{O}\left(r\sqrt{\frac{d}{m}(1 + r + r^2)}\right)$, i.e., looser by a factor of $\tilde{O}\left(\sqrt{1 + r + r^2}\right)$ compared to Cor. 15.

# 6 Discussion

We have presented high-probability PB bounds based on integral probability metrics, that extend standard PB bounds based on KL divergence and more recent $f$-divergence and $\alpha$-divergence based bounds, to a new class of distances. Our bounds interpolate between classic UC bounds and PB bounds, by allowing data- and algorithm-dependent complexity terms. As in all PB results, our bounds suggest improved rates when the PB posterior is close to the prior. While we have extended high-probability PB bounds for IPMs to novel distance measures, it is still an open question to do so without the UC assumption.

Possible directions for future research include: *(i)* Deriving high probability IPM-PB bounds (e.g. Wasserstein or TV based), without global UC assumptions (possibly using localization based approaches, e.g. , Local Rademacher complexities (Bartlett, Bousquet, & Mendelson, 2005; Koltchinskii & Panchenko, 2004) are computed only on a subset of hypotheses with small empirical risk). This may allow non-vacuous bounds for large-scale models where global UC-based bounds are extremely vacuous. *(ii)* Derivation of algorithms that utilize the bounds as minimization objectives. Based on the optimization advantages of Wasserstein based costs mentioned in Sec. 5.1, these could lead to enhanced practical utility. Such an advantage could play an important role in meta-learning schemes where PB methods have been widely used in recent years (Amit & Meir, 2018).

**Acknowledgments**

We thank Nadav Merlis and Daniel Soudry for helpful discussions of this work, and the anonymous reviewers for their helpful comments. Shay Moran is a Robert J. Shillman Fellow; he acknowledges support by ISF grant 1225/20, by BSF grant 2018385, by an Azrieli Faculty Fellowship, by Israel PBC-VATAT, by the Technion Center for Machine Learning and Intelligent Systems (MLIS), and by the the European Union (ERC, GENERALIZATION, 101039692). Views and opinions expressed

are however those of the author(s) only and do not necessarily reflect those of the European Union or the European Research Council Executive Agency. Neither the European Union nor the granting authority can be held responsible for them. The work of Ron Meir is partially supported by ISF grant 1693/22, by the Ollendorff Center of the Viterbi ECE Faculty at the Technion, and by the Skillman chair in biomedical sciences.

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
