# A  Appendix: Proofs

## A.1  Proof of the IPM-PB Bound (Prop. 4)

*Proof.* The proof follows a similar structure as the classical derivation (McAllester, 2003; Shalev-Shwartz & Ben-David, 2014), except for replacing the change-of-measure inequality.

For any sample $S \in \mathcal{Z}^m$, consider the function

$$f_S(h) \stackrel{\text{def}}{=} 2(m-1)\Delta_S^2(h).$$

Since we assume $f_S \in \mathcal{F}_S$, Definition 3 implies that for any pair of probability measures $P, Q \in \mathcal{M}(\mathcal{H})$

$$\mathbb{E}_{h \sim Q}[f_S(h)] - \mathbb{E}_{h \sim P}[f_S(h)] \leq \gamma_{\mathcal{F}_S}(Q, P).$$

Therefore, by the monotonicity of $\exp(\cdot)$ we have

$$\exp\left(\mathbb{E}_{h \sim Q}[f_S(h)] - \gamma_{\mathcal{F}_S}(Q, P)\right) \leq \exp\left(\mathbb{E}_{h \sim P}[f_S(h)]\right) \tag{24}$$

$$\leq \mathbb{E}_{h \sim P}[\exp(f_S(h))]. \tag{25}$$

where the last inequality is by the convexity of $\exp(\cdot)$, and by Jensen's inequality.

Taking the supremum over $Q \in \mathcal{M}(\mathcal{H})$, and an expectation over samples $S \sim \mathcal{D}^m$ we have that for any $P \in \mathcal{M}(\mathcal{H})$

$$\mathbb{E}_{S \sim \mathcal{D}^m} \sup_Q \left\{ \exp\left(\mathbb{E}_{h \sim Q}[f_S(h)] - \gamma_{\mathcal{F}_S}(Q, P)\right) \right\} \leq \mathbb{E}_{S \sim \mathcal{D}^m} \sup_Q \left\{ \mathbb{E}_{h \sim P} \exp(f_S(h)) \right\} \tag{26}$$

$$= \mathbb{E}_{S \sim \mathcal{D}^m} \mathbb{E}_{h \sim P} \exp(f_S(h))$$

$$= \mathbb{E}_{h \sim P} \mathbb{E}_{S \sim \mathcal{D}^m} \exp(f_S(h)),$$

where the last equality is obtained by the prior's independence from the sample, and from Fubini's theorem. We recall that by Hoeffding's inequality, for any $h \in \mathcal{H}$,

$$\mathbb{P}_{S \sim \mathcal{D}^m}\left(\Delta_S(h) > u\right) \leq e^{-2mu^2},$$

which, by Lemma 5 of McAllester (2003), this imply.

$$\mathbb{E}_{S \sim \mathcal{D}^m} \exp(f_S(h)) = \mathbb{E}_{S \sim \mathcal{D}^m} \exp\left(2(m-1)\Delta_S^2(h)\right) \leq m. \tag{27}$$

Inequalities (26) and (27) imply

$$\mathbb{E}_{S \sim \mathcal{D}^m} \sup_Q \left\{ \exp\left(\mathbb{E}_{h \sim Q}[f_S(h)] - \gamma_{\mathcal{F}_S}(Q, P)\right) \right\} \leq m.$$

Therefore, by Markov's inequality, for any $t > 0$ we have

$$\mathbb{P}_{S \sim \mathcal{D}^m}\left(\sup_Q \left\{ \exp\left(\mathbb{E}_{h \sim Q}[f_S(h)] - \gamma_{\mathcal{F}_S}(Q, P)\right) \right\} \geq t\right) \leq \frac{m}{t}.$$

Or, equivalently,

$$\mathbb{P}_{S \sim \mathcal{D}^m}\left(\ln\left(\sup_Q \left\{ \exp\left(\mathbb{E}_{h \sim Q}[f_S(h)] - \gamma_{\mathcal{F}_S}(Q, P)\right) \right\}\right) \geq \ln(t)\right) \leq \frac{m}{t}.$$

By Lem. 18, the $\ln(\cdot)$ and $\sup(\cdot)$ operations are interchangeable, and therefore

$$\mathbb{P}_{S \sim \mathcal{D}^m}\left(\sup_Q \left\{ \mathbb{E}_{h \sim Q}[f_S(h)] - \gamma_{\mathcal{F}_S}(Q, P)\right\} \geq \ln(t)\right) \leq \frac{m}{t}.$$

Let $\delta \in (0,1)$, we set $t = \frac{m}{\delta}$, and plug in $f_S(h) \stackrel{\text{def}}{=} 2(m-1)\Delta_S^2(h)$ to get

$$\mathbb{P}_{S \sim \mathcal{D}^m}\left(\sup_Q\left\{\mathop{\mathbb{E}}_{h \sim Q}\left(2(m-1)\Delta_S^2(h)\right) - \gamma_{\mathcal{F}_S}(Q,P)\right\} \geq \ln(m/\delta)\right) \leq \delta.$$

Therefore, the complementary event satisfies

$$\mathbb{P}_{S \sim \mathcal{D}^m}\left(\sup_Q\left\{\mathop{\mathbb{E}}_{h \sim Q}\left(2(m-1)\Delta_S^2(h)\right) - \gamma_{\mathcal{F}_S}(Q,P)\right\} < \ln(m/\delta)\right) \geq 1 - \delta.$$

I.e., for any $P \in \mathcal{M}(\mathcal{H})$, with a probability of at least $1 - \delta$ over the samples $S \sim \mathcal{D}^m$, the following inequality holds for all $Q \in \mathcal{M}(\mathcal{H})$

$$\mathop{\mathbb{E}}_{h \sim Q}\left(\Delta_S^2(h)\right) < \frac{\gamma_{\mathcal{F}_S}(Q,P) + \ln(m/\delta)}{2(m-1)}.$$

Jensen's inequality implies that

$$\left(\mathop{\mathbb{E}}_{h \sim Q}\Delta_S(h)\right)^2 \leq \mathop{\mathbb{E}}_{h \sim Q}\left(\Delta_S^2(h)\right) \leq \frac{\gamma_{\mathcal{F}}(Q,P) + \ln(m/\delta)}{2(m-1)}.$$

The proof is concluded by taking the square root of both sides. $\qquad\square$

## A.2 Proof of the Seeger's Type IPM-PB Bound (Prop. 5)

*Proof.* As in the proof of Prop. 4, we follow a similar structure as the classical derivation (Maurer, 2004; McAllester, 2003), except replacing the change-of-measure inequality.

For any sample $S \in \mathcal{Z}^m$, consider the function on $\mathcal{H}$

$$f_S(h) \stackrel{\text{def}}{=} m \cdot \mathtt{kl}(\hat{L}_S(h) \| L_D(h)).$$

Note that $f_S(\cdot)$ is almost surely well-defined, since if $L_D(h) = 0$, then $\hat{L}(h) \stackrel{a.s.}{=} 0$.

Since we assume $f_S \in \mathcal{F}_S$, Definition 3 implies that for any pair of probability measures $P, Q \in \mathcal{M}(\mathcal{H})$

$$\mathop{\mathbb{E}}_{h \sim Q}[f_S(h)] - \mathop{\mathbb{E}}_{h \sim P}[f_S(h)] \leq \gamma_{\mathcal{F}_S}(Q,P).$$

Therefore, by the monotonicity of $\exp(\cdot)$ we have

$$\exp\left(\mathop{\mathbb{E}}_{h \sim Q}[f_S(h)] - \gamma_{\mathcal{F}_S}(Q,P)\right) \leq \exp\left(\mathop{\mathbb{E}}_{h \sim P}[f_S(h)]\right)$$
$$\leq \mathop{\mathbb{E}}_{h \sim P}[\exp(f_S(h))].$$

where the last inequality is by the convexity of $\exp(\cdot)$ and by Jensen's inequality.

Taking the supremum over $Q \in \mathcal{M}(\mathcal{H})$, and an expectation over samples $S \sim \mathcal{D}^m$ we have for any $P \in \mathcal{M}(\mathcal{H})$

$$\mathbb{E}_{S \sim \mathcal{D}^m}\sup_Q\left\{\exp\left(\mathop{\mathbb{E}}_{h \sim Q}[f_S(h)] - \gamma_{\mathcal{F}_S}(Q,P)\right)\right\} \leq \mathbb{E}_{S \sim \mathcal{D}^m}\sup_Q\left\{\mathop{\mathbb{E}}_{h \sim P}\exp(f_S(h))\right\} \quad (28)$$
$$= \mathbb{E}_{S \sim \mathcal{D}^m}\mathop{\mathbb{E}}_{h \sim P}\exp(f_S(h))$$
$$= \mathop{\mathbb{E}}_{h \sim P}\mathbb{E}_{S \sim \mathcal{D}^m}\exp(f_S(h)),$$

where the last equality is obtained by the prior's independence from the sample, and from Fubini's theorem. Using Maurer (2004), Thm. 1 we have

$$\mathbb{E}_{S \sim \mathcal{D}^m}\exp(f_S(h)) = \mathbb{E}_{S \sim \mathcal{D}^m}\exp\left(m \cdot \mathtt{kl}(\hat{L}_S(h) \| L_D(h))\right) \quad (29)$$
$$\leq 2\sqrt{m}.$$

Inequalities (28) and (29) imply

$$\mathbb{E}_{S \sim \mathcal{D}^m} \sup_Q \left\{ \exp\left( \underset{h \sim Q}{\mathbb{E}} [f_S(h)] - \gamma_{\mathcal{F}_S}(Q, P) \right) \right\} \leq 2\sqrt{m}.$$

Therefore, by Markov's inequality, for any $t > 0$ we have

$$\mathbb{P}_{S \sim \mathcal{D}^m} \left( \sup_Q \left\{ \exp\left( \underset{h \sim Q}{\mathbb{E}} [f_S(h)] - \gamma_{\mathcal{F}_S}(Q, P) \right) \right\} \geq t \right) \leq \frac{2\sqrt{m}}{t},$$

or, equivalently,

$$\mathbb{P}_{S \sim \mathcal{D}^m} \left( \ln\left( \sup_Q \left\{ \exp\left( \underset{h \sim Q}{\mathbb{E}} [f_S(h)] - \gamma_{\mathcal{F}_S}(Q, P) \right) \right\} \right) \geq \ln(t) \right) \leq \frac{2\sqrt{m}}{t}.$$

By Lem. 18, the $\ln(\cdot)$ and $\sup(\cdot)$ operations are interchangeable, and therefore

Let $\delta \in (0, 1)$, we set $t = \frac{2\sqrt{m}}{\delta}$, and plug in $f_S(h) \stackrel{\text{def}}{=} m \cdot \mathtt{kl}(\hat{L}_S(h) \parallel L_D(h))$ to get

$$\mathbb{P}_{S \sim \mathcal{D}^m} \left( \sup_Q \left\{ m \underset{h \sim Q}{\mathbb{E}} \mathtt{kl}(\hat{L}_S(h) \parallel L_D(h)) - \gamma_{\mathcal{F}_S}(Q, P) \right\} \geq \ln(2\sqrt{m}/\delta) \right) \leq \delta.$$

Therefore, the complementary event satisfies

$$\mathbb{P}_{S \sim \mathcal{D}^m} \left( \sup_Q \left\{ m \underset{h \sim Q}{\mathbb{E}} \mathtt{kl}(\hat{L}_S(h) \parallel L_D(h)) - \gamma_{\mathcal{F}_S}(Q, P) \right\} < \ln(2\sqrt{m}/\delta) \right) \geq 1 - \delta.$$

I.e., for any $P \in \mathcal{M}(\mathcal{H})$, with a probability of at least $1 - \delta$ over the samples $S \sim \mathcal{D}^m$, the following inequality holds for all $Q \in \mathcal{M}(\mathcal{H})$

$$\underset{h \sim Q}{\mathbb{E}} \mathtt{kl}(\hat{L}_S(h) \parallel L_D(h)) < \frac{\gamma_{\mathcal{F}_S}(Q, P) + \ln(2\sqrt{m}/\delta)}{m}.$$

By the convexity of the function $\mathtt{kl}(p \parallel q)$ in the pair of parameters $(p, q)$ and by Jensen's inequality, we have

$$\mathtt{kl}(\underset{h \sim Q}{\mathbb{E}} \hat{L}_S(h) \parallel \underset{h \sim Q}{\mathbb{E}} L_D(h)) \leq \underset{h \sim Q}{\mathbb{E}} \mathtt{kl}(\hat{L}_S(h) \parallel L_D(h)).$$

Therefore we finally get that for any $P \in \mathcal{M}(\mathcal{H})$, w.p. of at least $1 - \delta$ the following holds for all $Q \in \mathcal{M}(\mathcal{H})$

$$\mathtt{kl}(\hat{L}_S(Q) \parallel L_D(Q)) < \frac{\gamma_{\mathcal{F}_S}(Q, P) + \ln(2\sqrt{m}/\delta)}{m}.$$

$\square$

### A.3   Proof of the Total-Variation PAC-Bayes Bound (Thm. 6)

*Proof.* Let $\delta > 0$. For any fixed sample $S \in \mathcal{Z}^m$, we define $f_S(h) \stackrel{\text{def}}{=} 2(m-1)\Delta_S^2(h)$. Define $M_S \stackrel{\text{def}}{=} \sup_{h \in \mathcal{H}} \Delta_S^2(h)$. Therefore, $0 \leq f_S(h) \leq 2(m-1)M_S$, i.e., $f_S(h) \in \mathcal{F}_{2(m-1)M_S}^\infty$. This fact, together with the general IPM PB bound (Prop. 4) and Eq. (10) (equivalence of IPM to TV under the family of bounded functions) imply that for any $\delta \in (0, 1)$

$$\mathbb{P}\left( \Delta_S(Q) \leq \sqrt{\frac{2(m-1)M_S D_{\mathrm{TV}}(Q, P)}{2(m-1)} + \frac{\ln(2m/\delta)}{2(m-1)}} \right) \geq 1 - \delta/2,$$

or equivalently,

$$\mathbb{P}\left( \Delta_S(Q) \leq \sqrt{M_S D_{\mathrm{TV}}(Q, P) + \frac{\ln(2m/\delta)}{2(m-1)}} \right) \geq 1 - \delta/2. \tag{30}$$

According to the UC assumption we have

$$\mathbb{P}(\Delta_S(h) \leq u(m, \delta/2), \forall h \in \mathcal{H}) \geq 1 - \delta/2,$$

and therefore, using the fact that $0 \leq \Delta_S(h) \leq 1$ we also have

$$\mathbb{P}\big(\Delta_S^2(h) \leq u^2(m, \delta/2), \forall h \in \mathcal{H}\big) \geq 1 - \delta/2,$$

which implies that the bounds also holds for the supremum $M_S$

$$\mathbb{P}\big(M_S \leq u^2(m, \delta/2)\big) \geq 1 - \delta/2. \tag{31}$$

To conclude the proof, we use a union bound argument and Equations (30) and (31).

$\square$

## A.4 Proof of the Template Wasserstein-PB Bound (Thm. 11)

*Proof.* Let $\delta > 0$. Let $K_S$ be some Lipschitz constant of $\Delta_S^2(\cdot)$. Define $f_S(h) \stackrel{\text{def}}{=} 2(m-1)K_S$. Notice that $\Delta_S^2(h)$ is $2(m-1)K_S$-Lipschitz, i.e., $f_S(h) \in \mathcal{F}_{2(m-1)K_S}^{\text{Lip}}$. Using Proposition 4 we have

$$\mathbb{P}\left(\Delta(Q) \leq \sqrt{\frac{\gamma_{\mathcal{F}_{2(m-1)K_S}^{\text{Lip}}}(P, Q) + \ln(2m/\delta)}{2(m-1)}}\right) \geq 1 - \delta/2.$$

By equation (17) (the Kantorovich-Rubinstein duality) the inequality can rewritten as

$$\mathbb{P}\left(\Delta(Q) \leq \sqrt{\frac{2(m-1)K_S W_1(Q, P) + \ln(2m/\delta)}{2(m-1)}}\right) \geq 1 - \delta/2. \tag{32}$$

By assumption, w.p. at least $1 - \delta/2$, $\Delta_S^2(h)$ is $K$-Lipschitz with $K = K(m, \delta/2)$. Using a union bound argument with this event and the event of (32) concludes the proof. $\square$

## A.5 Proof of the Wasserstein-PB Bound for Finite Classes (Thm. 12)

We first prove the following lemma.

**Lemma 16.** *Let $\mathcal{H}$ be a finite hypothesis class. Assume that for any fixed $z \in \mathcal{Z}$, $\ell(h, z)$ is a $G$-Lipschitz function in $h \in \mathcal{H}$ w.r.t the metric $\rho$. Then for any $\delta \in (0, 1)$, we have*

$$\mathbb{P}\left(\tilde{K}_S \leq \frac{8}{m} G \log(2|\mathcal{H}|/\delta)\right) \geq 1 - \delta,$$

*where for any fixed $S \in \mathcal{Z}^m$, $\tilde{K}_S$ is the sharp Lipschitz constant of $\Delta_S^2(\cdot)$, i.e.*

$$\tilde{K}_S \stackrel{\text{def}}{=} \sup_{h, h' \in \mathcal{H}: h \neq h'} \frac{\left|\Delta_S^2(h) - \Delta_S^2(h')\right|}{\rho(h, h')}.$$

*Proof of lemma 16.* Let $\delta > 0$.

Note that for any $h, h' \in \mathcal{H}$ and $S \in \mathcal{Z}^m$, we have

$$\frac{|\Delta_S(h) - \Delta_S(h')|}{\rho(h, h')} \tag{33}$$

$$= \frac{\left|\left[\mathbb{E}_{z \sim \mathcal{D}}\ell(h, z)) - \frac{1}{m}\sum_{i=1}^m \ell(h, z_i)\right] - \left[\mathbb{E}_{z \sim \mathcal{D}}\ell(h', z)) - \frac{1}{m}\sum_{i=1}^m \ell(h', z_i)\right]\right|}{\rho(h, h')}$$

$$= \left|\frac{\frac{1}{m}\sum_{i=1}^m [\ell(h', z_i) - \ell(h, z_i)] - \mathbb{E}_{z \sim \mathcal{D}}[\ell(h', z) - \ell(h, z)]}{\rho(h, h')}\right|.$$

By assumption, for any fixed $z \in \mathcal{Z}$, $\ell(h, z)$ is a $G$-Lipschitz function in $h$ w.r.t. the metric $\rho$. I.e., we have that $|\ell(h, z) - \ell(h', z)| \leq G\rho(h, h'), \forall h, h' \in \mathcal{H}, z \in \mathcal{Z}$. Hence, for any pair $(h, h') \in \mathcal{H}^2$, the random sequence $\{|\ell(h, z_i) - \ell(h', z_i)|\}_{i=1}^{m}$ is i.i.d. and bounded by $G\rho(h, h')$.

By Hoeffding's theorem, it holds with probability of at least $1 - \frac{\delta}{2|\mathcal{H}|^2}$ that

$$\left| \frac{1}{m} \sum_{i=1}^{m} [\ell(h', z_i) - \ell(h, z_i)] - \mathbb{E}_{z \sim \mathcal{D}}[\ell(h', z) - \ell(h, z)] \right| \leq \rho(h, h') G \sqrt{\frac{2 \ln(\frac{4|\mathcal{H}|^2}{\delta})}{m}}. \tag{34}$$

By using a union bound over claim (34) for all pairs of hypotheses $(h, h') \in \mathcal{H}^2$, we get that w.p. of at least $1 - \delta/2$, we have for all pairs $(h, h') \in \mathcal{H}^2$ **simultaneously** that,

$$\left| \frac{1}{m} \sum_{i=1}^{m} [\ell(h', z_i) - \ell(h, z_i)] - \mathbb{E}_{z \sim \mathcal{D}}[\ell(h', z) - \ell(h, z)] \right| \leq \rho(h, h') G \sqrt{\frac{2 \ln(\frac{4|\mathcal{H}|^2}{\delta})}{m}}. \tag{35}$$

It is well-known (e.g. Shalev-Shwartz and Ben-David (2014), Cor. 2.3) that for a finite hypothesis class and any $\delta/2 > 0$,

$$\mathbb{P}\left( \sup_{h \in \mathcal{H}} |\Delta_S(h)| \leq \sqrt{\frac{\ln(4|\mathcal{H}|/\delta)}{2m}} \right) \geq 1 - \delta/2. \tag{36}$$

Notice that

$$\begin{aligned}
\tilde{K}_S &\overset{\text{def}}{=} \sup_{h, h' \in \mathcal{H}: h \neq h'} \frac{\left| \Delta_S^2(h) - \Delta_S^2(h') \right|}{\rho(h, h')} \\
&= \sup_{h, h' \in \mathcal{H}: h \neq h'} \frac{|\Delta_S(h) - \Delta_S(h')||\Delta_S(h) + \Delta_S(h')|}{\rho(h, h')} \\
&\leq \sup_{h, h' \in \mathcal{H}: h \neq h'} \frac{|\Delta_S(h) - \Delta_S(h')|}{\rho(h, h')} 2 \sup_{h'' \in \mathcal{H}} |\Delta_S(h'')| \\
&\overset{(33)}{=} 4 \sup_{h, h' \in \mathcal{H}: h \neq h'} \left| \frac{\frac{1}{m} \sum_{i=1}^{m} [\ell(h', z_i) - \ell(h, z_i)] - \mathbb{E}_{z \sim \mathcal{D}}[\ell(h', z) - \ell(h, z)]}{\rho(h, h')} \right| \sup_{h'' \in \mathcal{H}} |\Delta_S(h'')|.
\end{aligned}$$

The proof of the lemma is concluded by using a union bound argument with claims (35) and (36). We get that w.p. of at least $1 - \delta$ we have

$$\begin{aligned}
\tilde{K}_S &\leq 4 \sup_{h, h' \in \mathcal{H}: h \neq h'} \left\{ \frac{\rho(h, h') G \sqrt{\frac{2 \ln(4|\mathcal{H}|^2/\delta)}{m}}}{\rho(h, h')} \right\} \sqrt{\frac{\ln(4|\mathcal{H}|/\delta)}{2m}} \\
&= \frac{4}{m} G \sqrt{\ln(4|\mathcal{H}|^2/\delta) \ln(4|\mathcal{H}|/\delta)} \\
&\leq \frac{4}{m} G \ln(4|\mathcal{H}|^2/\delta) \\
&\leq \frac{8}{m} G \ln(2|\mathcal{H}|/\delta).
\end{aligned}$$

$\square$

*Proof of Theorem 12.* The proof follows directly from Lemma 16 and Theorem 11. $\square$

### A.6 Proof of the Wasserstein-PB Bound for Differentiable Loss UC Classes (Thm. 14)

We first prove the following lemma.

**Lemma 17.** *Let $(\mathcal{H}, \rho)$ be a metric space such that $\mathcal{H} \subset \mathbb{R}^d$ is a closed convex set, and $\rho$ is the $L_2$ distance. Assume that the loss function $\ell(h, z)$ is differentiable w.r.t. h on Int$(\mathcal{H}) \times \mathcal{Z}$ and continuous w.r.t. h on $\mathcal{H} \times \mathcal{Z}$. Assume the learning problem has a UC bound $u$ (Def. 1), and a UC bound of the loss gradient, $u^{grad}$ (Def. 13), then*

$$\mathbb{P}\Big( \tilde{K}_S \leq 2 \cdot u(m, \delta/2) \cdot u^{grad}(m, \delta/2) \Big) \geq 1 - \delta,$$

*where for any fixed $S \in \mathcal{Z}^m$, $\tilde{K}_S$ is the sharp Lipschitz constant of $\Delta_S^2(\cdot)$, i.e.*

$$\tilde{K}_S \stackrel{def}{=} \sup_{h, h' \in \mathcal{H} : h \neq h'} \frac{\left| \Delta_S^2(h) - \Delta_S^2(h') \right|}{\rho(h, h')}.$$

*Proof of Lemma 17.* Using the fact that loss function $\ell(h, z)$ is differentiable w.r.t. h on Int$(\mathcal{H}) \times \mathcal{Z}$ and continuous w.r.t. h on $\mathcal{H} \times \mathcal{Z}$, and by the mean value theorem and the convexity of $\mathcal{H}$, we have

$$\forall z \in \mathcal{Z}, (h, h') \in \mathcal{H}^2, \exists w_{z,h,h'} \in \mathcal{H}, \text{s.t. } \ell(h, z) - \ell(h', z) = \langle h - h', \nabla_h \ell(w_{z,h,h'}, z) \rangle, \quad (37)$$

where $\nabla_h \ell(w, z)$ denotes the gradient of $\ell(\cdot, \cdot)$ w.r.t the h variable, at the point $(w, z)$.

Notice that

$$\tilde{K}_S \stackrel{def}{=} \sup_{h, h' \in \mathcal{H} : h \neq h'} \frac{\left| \Delta_S^2(h) - \Delta_S^2(h') \right|}{\rho(h, h')} \qquad (38)$$

$$= \sup_{h, h' \in \mathcal{H} : h \neq h'} \frac{|\Delta_S(h) - \Delta_S(h')||\Delta_S(h) + \Delta_S(h')|}{\rho(h, h')}$$

$$\leq \sup_{h, h' \in \mathcal{H} : h \neq h'} \frac{|\Delta_S(h) - \Delta_S(h')|}{\rho(h, h')} 2 \sup_{h'' \in \mathcal{H}} |\Delta_S(h'')|.$$

We have for any $(h, h') \in \mathcal{H}, h \neq h$ that

$$\frac{|\Delta_S(h) - \Delta_S(h')|}{\rho(h, h')} \qquad (39)$$

$$= \left| \frac{\frac{1}{m} \sum_{i=1}^m [\ell(h', z_i) - \ell(h, z_i)] - \mathbb{E}_{z \sim \mathcal{D}}[\ell(h', z) - \ell(h, z)]}{\rho(h, h')} \right|$$

$$\stackrel{(i)}{=} \left| \frac{\frac{1}{m} \sum_{i=1}^m \langle h - h', \nabla_h \ell(w_{z_i,h,h'}, z_i) \rangle - \mathbb{E}_{z \sim \mathcal{D}} \langle h - h', \nabla_h \ell(w_{z,h,h'}, z) \rangle}{\|h - h'\|_2} \right|$$

$$\stackrel{(ii)}{=} \left| \frac{\langle h - h', \frac{1}{m} \sum_{i=1}^m \nabla_h \ell(w_{z_i,h,h'}, z_i) - \mathbb{E}_{z \sim \mathcal{D}} \nabla_h \ell(w_{z,h,h'}, z) \rangle}{\|h - h'\|_2} \right|$$

$$\stackrel{(iii)}{\leq} \frac{\|h - h'\|_2 \left\| \frac{1}{m} \sum_{i=1}^m \nabla_h \ell(w_{z_i,h,h}, z_i) - \mathbb{E}_{z \sim \mathcal{D}} \nabla_h \ell(w_{z,h,h}, z) \right\|_2}{\|h - h'\|_2}$$

$$= \left\| \frac{1}{m} \sum_{i=1}^m \nabla_h \ell(w_{z_i,h,h'}, z_i) - \mathbb{E}_{z \sim \mathcal{D}} \nabla_h \ell(w_{z,h,h'}, z) \right\|_2,$$

where *(i)* is by the mean value theorem (Eq. 37), *(ii)* is by the linearity of the sum, expectation, and the inner product, and *(iii)* is by the Cauchy–Schwarz inequality.

By the UC assumptions, we have

$$\mathbb{P}(\forall h \in \mathcal{H}, |\Delta_S(h)| \leq u(m, \delta/2)) \geq 1 - \delta/2, \qquad (40)$$

and

$$\mathbb{P}\left( \forall h \in \text{Int}(\mathcal{H}), \left\| \frac{1}{m} \sum_{i=1}^m \nabla_h \ell(h, z_i) - \mathbb{E}_{z \sim \mathcal{D}} \nabla_h \ell(h, z) \right\|_2 \leq u^{grad}(m, \delta/2) \right) \geq 1 - \delta/2. \qquad (41)$$

To conclude the proof, we use a union bound argument with (40) and (41), and use inequalities (38) and (39) to finally get

$$\mathbb{P}\Big( \tilde{K}_S \leq 2 \cdot u(m, \delta/2) \cdot u^{grad}(m, \delta/2) \Big) \geq 1 - \delta.$$

$\square$

*Proof of Theorem 14.* The proof follows from Lemma 17 with $\delta/2$, Theorem 11 with $\delta/2$, and using the union bound. $\square$

## A.7 Proof of the Wasserstein-PB Bound for Linear Regression (Cor. 15)

*Proof of Corollary 15.* To meet the requirements of Theorem 14, we will prove a uniform convergence bound for the generalization gap (Def. 1), and for the loss gradient (Def. 13).

The generalization gap function for any $h \in \mathcal{H}$ can be written as

$$\Delta_S(h) = \mathbb{E}\ell(h, z) - \frac{1}{m}\sum_{i=1}^{m}\ell(h, z_i) \tag{42}$$

$$= \mathbb{E}\frac{1}{4}(h^\top x - y)^2 - \frac{1}{m}\sum_{i=1}^{m}\frac{1}{4}(h^\top x_i - y_i)^2$$

$$= \frac{1}{4}\left(\mathbb{E}y^2 - \frac{1}{m}\sum_{i=1}^{m}y_i^2\right) - \frac{1}{2}\left(\mathbb{E}yx^\top h - \frac{1}{m}\sum_{i=1}^{m}y_i x_i^\top h\right) + \frac{1}{4}h^\top\left(\mathbb{E}xx^\top - \frac{1}{m}\sum_{i=1}^{m}x_i x_i^\top\right)h.$$

Let $\delta > 0$. We will now bound in high probability each of the three term above.

**First term.** The variables $y_1^2, \ldots, y_m^2$ are independent random variables in the range $[0, 1]$, therefore by Hoeffding's inequality

$$\mathbb{P}\left(\left|\mathbb{E}y^2 - \frac{1}{m}\sum_{i=1}^{m}y_i^2\right| \leq \sqrt{\frac{\ln(6/\delta)}{2m}}\right) \geq 1 - \delta/3. \tag{43}$$

I.e.,

$$\mathbb{P}\left(\frac{1}{4}\left|\mathbb{E}y^2 - \frac{1}{m}\sum_{i=1}^{m}y_i^2\right| \leq \sqrt{\frac{\ln(6/\delta)}{32m}}\right) \geq 1 - \delta/3. \tag{44}$$

**Second term.** Note that $yx$ is $r$-sub-Gaussian random vector in $\mathbb{R}^d$, since, for any $s$ in the unit-sphere $\mathbb{S}_1^{d-1}$, $s^\top yx$ is $r$-sub-Gaussian (since it is a.s. bounded in $[-r, r]$). Therefore $\{y_i x_i\}_{i=1}^{m}$ are independent $r$-sub-Gaussian random vectors. Using Thm. 1 of Hsu, Kakade, and Zhang (2012), we have that for any $t > 0$,

$$\mathbb{P}\left(\left\|\mathbb{E}yx - \frac{1}{m}\sum_{i=1}^{m}y_i x_i\right\|_2^2 > \frac{r^2}{m}\left(d + 2d\sqrt{t} + 2t\right)\right) \leq \exp(-t).$$

Therefore, we have

$$\mathbb{P}\left(\left\|\mathbb{E}yx - \frac{1}{m}\sum_{i=1}^{m}y_i x_i\right\|_2^2 < \frac{r^2}{m}\left(d + 2d\sqrt{\ln(3/\delta)} + 2\ln(3/\delta)\right)\right) \geq 1 - \delta/3. \tag{45}$$

Then, by the Cauchy–Schwarz inequality we have that w.p. of at least $1 - \delta/3$, for all $h \in \mathbb{B}_{1/r}^d$

$$\frac{1}{2}\left|h^\top\left(\mathbb{E}yx^\top - \frac{1}{m}\sum_{i=1}^{m}y_i x_i^\top\right)\right| \leq \frac{1}{2}\|h\|_2\left\|\mathbb{E}yx - \frac{1}{m}\sum_{i=1}^{m}y_i x_i\right\|_2 \tag{46}$$

$$< \frac{1}{2\sqrt{m}}\sqrt{d + 2d\sqrt{\ln(3/\delta)} + 2\ln(3/\delta)}.$$

**Third term.** By Theorem 6.5 of Wainwright (2019) (constants from Thm. of Bastani, Simchi-Levi, and Zhu (2021) Lem. 22), we have that w.p. of at least $1 - \delta/3$

$$\frac{1}{4}\left\|\frac{1}{m}\sum_{i=1}^{m}x_i x_i^\top - \mathbb{E}\left(xx^\top\right)\right\|_{op} \leq 8r^2\max\left\{\sqrt{\frac{5d + 2\ln\left(\frac{6}{\delta}\right)}{m}}, \frac{5d + 2\ln\left(\frac{6}{\delta}\right)}{m}\right\}, \tag{47}$$

where $\|\cdot\|_{\mathrm{op}}$ is the $\ell_2$ operator-norm, that can be defined by $\|A\|_{\mathrm{op}} \stackrel{\text{def}}{=} \sup_{u,v\in\mathbb{S}_1^{d-1}} |u^\top A v|, \forall A \in \mathbb{R}^{d\times d}$, where $\mathbb{S}_1^{d-1}$ is the the unit sphere in $\mathbb{R}^d$. Therefore, we conclude that w.p. of at least $1 - \delta/3$

$$\forall h \in \mathbb{B}_{1/r}^d, \frac{1}{4}\left|h^\top\left(\frac{1}{m}\sum_{i=1}^m x_i x_i^\top - \mathbb{E}(xx^\top)\right)h\right| \leq 8\max\left\{\sqrt{\frac{5d + 2\ln\left(\frac{6}{\delta}\right)}{m}}, \frac{5d + 2\ln\left(\frac{6}{\delta}\right)}{m}\right\}. \tag{48}$$

Taking the absolute value of (42) and using the triangle inequality, the union bound, and inequalities (44), (46), and (48), we get that

$$\mathbb{P}(\forall h \in \mathcal{H}, \Delta_S(h) \leq u(m,\delta)) \geq 1 - \delta,$$

where we defined

$$u(m,\delta) \stackrel{\text{def}}{=} \sqrt{\frac{\ln(6/\delta)}{32m}} + \sqrt{\frac{d + 2d\sqrt{\ln(3/\delta)} + 2\ln(3/\delta)}{4m}} \tag{49}$$

$$+ 8\max\left\{\sqrt{\frac{5d + 2\ln\left(\frac{6}{\delta}\right)}{m}}, \frac{5d + 2\ln\left(\frac{6}{\delta}\right)}{m}\right\}.$$

Therefore, $u(m,\delta) \in O\left(\sqrt{\frac{d(1+\ln(1/\delta))}{m}}\right)$.

Next, we wish to prove a uniform convergence bound for the loss gradient, in Euclidean norm. Note that for any $h \in \mathcal{H}$

$$\mathbb{E}_{z\sim\mathcal{D}}\nabla_h\ell(h,z) - \frac{1}{m}\sum_{i=1}^m \nabla_h\ell(h,z_i) \tag{50}$$

$$= \mathbb{E}_{z\sim\mathcal{D}}\nabla_h\frac{1}{4}(h^\top x - y)^2 - \frac{1}{m}\sum_{i=1}^m \nabla_h\frac{1}{4}(h^\top x_i - y_i)^2$$

$$= \frac{1}{2}\mathbb{E}(h^\top x - y)x^\top - \frac{1}{m}\sum_{i=1}^m \frac{1}{2}(h^\top x_i - y_i)x_i^\top$$

$$= \frac{1}{2}h^\top\left(\mathbb{E}xx^\top - \frac{1}{m}\sum_{i=1}^m x_i x_i^\top\right) - \frac{1}{2}\left(\mathbb{E}yx^\top - \frac{1}{m}\sum_{i=1}^m y_i x_i^\top\right).$$

To bound the $L_2$ norm of the first term of the equation above, we use similar argument as in (47), and the fact that the operator-norm can be defined equivalently by $\forall A \in \mathbb{R}^{d\times d}, \|A\|_{\mathrm{op}} = \sup_{v\in\mathbb{S}_1^{d-1}}\|Av\|_2$, to get that w.p. of at least $1 - \delta/2$

$$\forall h \in \mathbb{B}_{1/r}^d, \left\|\frac{1}{2}h^\top\left(\frac{1}{m}\sum_{i=1}^m x_i x_i^\top - \mathbb{E}(xx^\top)\right)\right\|_2 \leq 16r\max\left\{\sqrt{\frac{5d + 2\ln\left(\frac{4}{\delta}\right)}{m}}, \frac{5d + 2\ln\left(\frac{4}{\delta}\right)}{m}\right\}. \tag{51}$$

To bound the $L_2$ norm of the second term, we use the same argument as in (45), and get

$$\mathbb{P}\left(\frac{1}{2}\left\|\mathbb{E}yx - \frac{1}{m}\sum_{i=1}^m y_i x_i\right\|_2 < \frac{r}{2\sqrt{m}}\sqrt{d + 2d\sqrt{\ln(2/\delta)} + 2\ln(2/\delta)}\right) \geq 1 - \delta/2. \tag{52}$$

Now, taking the norm of equality (50) and using the triangle inequality, inequalities (51) and (52), and the union bound we get that

$$\mathbb{P}\left(\forall h \in \mathcal{H}, \left\|\mathbb{E}_{z\sim\mathcal{D}}\nabla_h\ell(h,z) - \frac{1}{m}\sum_{i=1}^m \nabla_h\ell(h,z_i)\right\|_2 \leq u^{\mathrm{grad}}(m,\delta)\right) \geq 1 - \delta, \tag{53}$$

where we defined

$$u^{\mathrm{grad}}(m, \delta) \stackrel{\mathrm{def}}{=} 16r \max \left\{ \sqrt{\frac{5d + 2\ln\left(\frac{4}{\delta}\right)}{m}}, \frac{5d + 2\ln\left(\frac{4}{\delta}\right)}{m} \right\} + r\sqrt{\frac{d + 2d\sqrt{\ln(2/\delta)} + 2\ln(2/\delta)}{4m}}.$$

(54)

Therefore, $u^{\mathrm{grad}}(m, \delta) \in O\left( r\sqrt{\frac{d(1 + \ln(1/\delta))}{m}} \right)$.

Notice that the loss is bounded in $[0, 1]$, since

$$\ell(x, y, h) = \frac{1}{4}(h^\top x - y)^2 \leq \frac{1}{4} 2((h^\top x)^2 + y^2) \leq \frac{1}{2}\left( \|h\|_2^2 \|x\|_2^2 + 1 \right) \leq 1.$$

Therefore we can use Theorem 14 to conclude the proof. $\qquad\square$

# B  Appendix: An Example of a Seeger Type Bound

To derive an analogous Seeger's type theorem to Thm. 6, we need to prove uniform convergence of the kl-gap, $\Delta_S^{\mathrm{kl}}(h) \stackrel{\mathrm{def}}{=} \mathrm{kl}(\hat{L}_S(h) \,\|\, L_D(h))$, rather than the usual gap $\Delta_S(h) = L_D(h) - \hat{L}_S(h)$.

For example, consider the binary classification and finite $\mathcal{H}$ case.

For each $h \in \mathcal{H}$, we bound $\Delta_S^{\mathrm{kl}}(h) = \mathrm{kl}(\hat{L}_S(h) \,\|\, L_D(h))$ using the concentration inequality from Dembo and Zeitouni (2009) Thm. 2.2.3. (see also Mardia, Jiao, Tánczos, Nowak, and Weissman (2019) Lem. 8), which holds since $\hat{L}_S(h)$ is an empirical average of $m$ Bernoulli i.i.d variables with mean $L_D(h)$. For any $\varepsilon > 0$, we have

$$\mathbb{P}\left(\Delta_S^{\mathrm{kl}}(h) < \varepsilon\right) \geq 1 - 2\exp(-m\varepsilon).$$

Using a union bound argument we get

$$\mathbb{P}\left(\forall h \in \mathcal{H}, \Delta_S^{\mathrm{kl}}(h) < \varepsilon\right) \geq 1 - 2|\mathcal{H}|\exp(-m\varepsilon).$$

Therefore, for any $\delta \in (0, 1)$ we can get

$$\mathbb{P}\left( \forall h \in \mathcal{H}, \Delta_S^{\mathrm{kl}}(h) < \frac{\ln(2|\mathcal{H}|/\delta)}{m} \right) \geq 1 - \delta.$$

(55)

Let $\mathcal{F}_{\ln(4|\mathcal{H}|/\delta)}^\infty$ be the family of functions as defined in Eq. 9, i.e., functions that are bounded in the $\infty$-norm by $\ln(4|\mathcal{H}|/\delta)$, for which the IPM is $\ln(4|\mathcal{H}|/\delta)D_{\mathrm{TV}}(Q, P)$.

By (55), w.p. at least $1 - \delta/2$ we have that $m\Delta_S^{\mathrm{kl}}(h) \in \mathcal{F}_{\ln(4|\mathcal{H}|/\delta)}^\infty$

Now we can use the Seeger's type IPM-PB bound (Prop. 5) and a union bound argument, to get that with probability at least $1 - \delta$ over the samples $S \sim \mathcal{D}^m$, the following inequality holds for all $Q \in \mathcal{M}(\mathcal{H})$

$$\Delta_S(Q) \leq \sqrt{2\hat{L}_S(Q)\frac{\ln(4|\mathcal{H}|/\delta)D_{\mathrm{TV}}(Q, P) + \ln(4\sqrt{m}/\delta)}{m}} + 2\frac{\ln(4|\mathcal{H}|/\delta)D_{\mathrm{TV}}(Q, P) + \ln(4\sqrt{m}/\delta)}{m}.$$

# C  Appendix: Numerical Demonstration Details

This section describes the experiment that implements the setting of Corollary 15 (Wasserstein-PB Bound for Linear Regression). The code is available at: `https://github.com/ron-amit/pac_bayes_reg`.

**The sample distribution.** The unknown data distribution $\mathcal{D}$ is determined by a latent vector $g \in \mathbb{R}^d$, drawn once per experiment instance from a uniform distribution over $\mathbb{B}_{0.1}$. The dimension

is $d = 10$. For each sample $(x, y) \sim \mathcal{D}$, $x$ is drawn uniformly from $\mathbb{B}_{0.1}$ and $y = f(x)$ is set by $f(x) = \text{clip}_{[-1,1]}\{g^\top x + \xi\}$ where,

$$\text{clip}_{[a,b]}(t) \stackrel{\text{def}}{=} \begin{cases} a, & t < a \\ t, & a \le t \le b \\ b & t > b, \end{cases}$$

for any $a, b, t \in \mathbb{R}$, and $\xi$ is drawn uniformly from $[-0.5, 0.5]$. The motivation for this choice of $\mathcal{D}$ is to have an underlying linear structure in the data, corrupted by noise. The clipping ensures that the loss values are in the range $[0, 1]$.

**The prior and posterior distributions.**   The hypothesis space is an $r$-radius ball $\mathcal{H} = \mathbb{B}_r$, with $r = 1$. The prior and posterior distributions over $\mathcal{H}$ are set as projected Gaussian distributions. Let $P_\mathbb{B} : \mathbb{R}^d \to \mathbb{B}_r$ be a projection operator onto $\mathbb{B}_r$. Let $\tilde{P}$ be a Gaussian measure over $\mathbb{R}^d$, $\mathcal{N}(\mu_P, \sigma_P^2 I)$, where $\mu_P = \mathbf{0}$, and $\sigma_P$ is a fixed constant that will be specified later. The prior is defined as $P = P_\mathbb{B} \sharp \tilde{P}$, i.e. , as the push-forward measure of $\tilde{P}$ under the projection $P_\mathbb{B}$. The family of posteriors we are considering are projected Gaussian distributions, $\mathcal{Q} \stackrel{\text{def}}{=} \left\{ P_\mathbb{B} \sharp \tilde{Q} : \tilde{Q} = \mathcal{N}(\mu_Q, \sigma_Q^2 I), \mu_Q, \in \mathbb{B}_{r_Q} \right\}$, where $\sigma_Q$ is a fixed constant that will be specified later and the maximal norm of $\mu_Q$ is $r_Q = 0.05$.

**The Wasserstein distance.**  Since there is no closed-form formula for the $1^{\text{st}}$ order Wasserstein distance between Gaussian distributions projected onto a ball $W_1(Q, P)$, we will instead use an upper bound. We use Lemma 19 (Sect. D) to bound this distance with the distance of the corresponding pre-projection measures, $W_1(\tilde{Q}, \tilde{P})$, where $\tilde{Q}$ and $\tilde{P}$ are the corresponding pre-projection measures. Note that our choice of parameters ensures that the lemma condition holds:

$$r^2 \ge \max\left\{ \|\mu_Q\|_2^2 + \|\Sigma_Q\|_F^2, \|\mu_P\|_2^2 + \|\Sigma_P\|_F^2 \right\} = \max\left\{ \|\mu_Q\|_2^2 + d\sigma_Q^2, \|\mu_P\|_2^2 + d\sigma_P^2 \right\}.$$

We also use the fact that $W_1(\tilde{Q}, \tilde{P}) \le W_2(\tilde{Q}, \tilde{P})$ ( Givens and Shortt (1984), Prop. 3) and the analytic formula for the $2^{\text{nd}}$ order Wasserstein distance between two Gaussian distributions (Givens and Shortt (1984), Prop. 7) to finally get a closed-form upper bound,

$$W_1(Q, P) \stackrel{\text{Lem. 19}}{\le} \sqrt{\|\mu_Q - \mu_P\|_2^2 + \mathbf{Tr}\left( \Sigma_Q + \Sigma_P - 2\left( \Sigma_Q^{1/2} \Sigma_P \Sigma_Q^{1/2} \right)^{1/2} \right)} \tag{56}$$

$$+ \sqrt{\frac{\pi}{2} \|\Sigma_Q\|_{2,2}} \, \text{erfc}\left( \frac{r - \sqrt{\|\mu_Q\|_2^2 + \|\Sigma_Q\|_F^2}}{\sqrt{2\|\Sigma_Q\|_{2,2}}} \right)$$

$$+ \sqrt{\frac{\pi}{2} \|\Sigma_P\|_{2,2}} \, \text{erfc}\left( \frac{r - \sqrt{\|\mu_P\|_2^2 + \|\Sigma_P\|_F^2}}{\sqrt{2\|\Sigma_P\|_{2,2}}} \right)$$

$$= \sqrt{\|\mu_Q - \mu_P\|_2^2 + d(\sigma_Q - \sigma_P)^2}$$

$$+ \sqrt{\frac{\pi}{2}} \sigma_Q \, \text{erfc}\left( \frac{r - \sqrt{\|\mu_Q\|_2^2 + d\sigma_Q^2}}{\sqrt{2}\sigma_Q} \right) + \sqrt{\frac{\pi}{2}} \sigma_P \, \text{erfc}\left( \frac{r - \sqrt{\|\mu_P\|_2^2 + d\sigma_P^2}}{\sqrt{2}\sigma_P} \right)$$

$$\stackrel{\text{def}}{=} W_{\text{bound}}(\mu_Q).$$

Notice that in the limit of $\sigma_Q, \sigma_P \to 0$, the bound becomes $\|\mu_Q - \mu_P\|_2$, which is equivalent to the Wasserstein distance between two Dirac measures at $\mu_Q$ and $\mu_P$.

**The empirical risk term.**   To compute the expectation of the empirical risk w.r.t. the posterior, $\mathbb{E}_{h \sim Q} \hat{L}(h)$, we derive a closed-form formula using the structure and of the loss and the posterior distribution [7] . Given a dataset $S = \{(x_i, y_i)\}_{i=1}^m$, denote $X \in \mathbb{R}^{m \times d}$ as a matrix whose rows are the

---

[7]In cases where the loss is a more complicated function (but still differentiable), one can approximate the expectation over the posterior with the reparametrization trick (D. P. Kingma & Welling, 2013), similarly to Amit and Meir (2018); Dziugaite and Roy (2017).

vectors $x_i$, and denote $Y \in \mathbb{R}^{m \times 1}$ as a vector whose entries are $y_i$. Denote $\hat{J}_{(X,Y)}(\mu_Q) \overset{\text{def}}{=} \mathbb{E}_{h \sim Q} \hat{L}(h)$. Then we have

$$\hat{J}_{(X,Y)}(\mu_Q) = \mathbb{E}_{h \sim \mathcal{N}(\mu_Q, \sigma_Q^2 I)} \frac{1}{m} \sum_{i=1}^{m} \frac{1}{4}(h^\top x_i - y_i)^2 \tag{57}$$

$$= \frac{1}{4m} \mathbb{E}_{h \sim \mathcal{N}(\mu_Q, \sigma_Q^2 I)} \left\| X h^\top - Y \right\|_2^2$$

$$= \frac{1}{4m} \mathbb{E}_{\epsilon \sim \mathcal{N}(0, I)} \left\| X(\mu_Q + \sigma_Q \epsilon)^\top - Y \right\|_2^2$$

$$= \frac{1}{4m} \mathbb{E}_{\epsilon \sim \mathcal{N}(0, I)} \left\| \sigma_Q X \epsilon^\top + X \mu_Q^\top - Y \right\|_2^2$$

$$= \frac{1}{4m} \left( \left\| X \mu_Q^\top - Y \right\|_2^2 + \mathbb{E}_{\epsilon \sim \mathcal{N}(0, I)} \sigma_Q^2 \, \mathbf{Tr}(\epsilon X^\top X \epsilon^\top) \right)$$

$$= \frac{1}{4m} \left( \left\| X \mu_Q^\top - Y \right\|_2^2 + \sigma_Q^2 \, \mathbf{Tr}(X^\top X) \right)$$

$$= \frac{1}{4m} \left( \left\| X \mu_Q^\top - Y \right\|_2^2 + \sigma_Q^2 \left\| X \right\|_F^2 \right).$$

**The explicit Wasserstein-PB bound.** According to Cor. 15, given a training set $S = (X, Y)$, the upper bound on the expected risk $L(Q) \overset{\text{def}}{=} \mathbb{E}_{h \sim Q} L(h)$ is

$$J_{(X,Y)}^{\text{WPB}}(\mu_Q) \overset{\text{def}}{=} \hat{J}_{(X,Y)}(\mu_Q) + \sqrt{2u(m, \delta/4) \cdot u^{\text{grad}}(m, \delta/4) \cdot W_{\text{bound}}(\mu_Q) + \frac{\ln(2m/\delta)}{2(m-1)}}, \tag{58}$$

where $u(m, \delta)$ is defined in (49), $u^{\text{grad}}(m, \delta)$ is defined in (54), and $W_{\text{bound}}(\mu_Q)$ is the upper bound over $W_1(Q, P)$ defined in (56).

**The explicit KL-PB bound.** For the KL-PB bound, we use the classic PB bound (Prop. 2) with the KL-divergence replaced by an upper bound that has a closed-form expression. By the data-processing inequality we have that $\text{KL}(Q \parallel P) = \text{KL}(P_\mathbb{B} \sharp \tilde{Q} \parallel P_\mathbb{B} \sharp \tilde{P}) \leq \text{KL}(\tilde{Q} \parallel \tilde{P})$, and $\text{KL}(\tilde{Q} \parallel \tilde{P})$ can be computed using the analytic formula for the KL-divergence between two Gaussian distributions. Therefore, the upper bound we use is

$$J_{(X,Y)}^{\text{KL-PB}}(\mu_Q) \overset{\text{def}}{=} \hat{J}_{(X,Y)}(\mu_Q) + \sqrt{\frac{\frac{\|\mu_Q - \mu_P\|_2^2}{2\sigma_P^2} + d\left(\ln\left(\frac{\sigma_P}{\sigma_Q}\right) + \frac{\sigma_Q^2}{2\sigma_P^2} - \frac{1}{2}\right) + \ln(m/\delta)}{2(m-1)}}. \tag{59}$$

**Experiment Procedure:** We repeat the experiment for 10 repetitions, to account for the randomness of the data and optimization in each run. In each run, *(i)* the task data distribution $\mathcal{D}$ is generated as described above, *(ii)* A training set of $m$ samples is generated. *(iii)* The posterior mean vector $\mu_Q$ is learned using the Adam Optimizer (D. Kingma & Ba, 2015) that minimizes either $J_{(X,Y)}^{\text{KL-PB}}(\mu_Q)$ or $J_{(X,Y)}^{\text{WPB}}(\mu_Q)$ (as will be specified later), where the learning rate is set as $10^{-3}$, and the maximal batch size is 256. The gradients are computed using automatic differentiation by the PyTorch framework (Paszke et al., 2019). After each gradient step, the parameter $\mu_Q$ is projected to $\mathbb{B}_{r_Q}$.

**Results.** Table 1 show the results when we set the prior parameter $\sigma_P$ as $10^{-2}$, and Table 2 shows the results for $\sigma_P = 10^{-4}$, both use $\sigma_Q = 10^{-3}$. The optimization objective for those two setups is the KLPB bound, $J_{(X,Y)}^{\text{KL-PB}}(\mu_Q)$.

The third setup (Table 3) investigates Dirac posteriors ("a deterministic model"). In this setup we set $\sigma_Q = \sigma_P = 0$, and the optimization objective is set to be $J_{(X,Y)}^{\text{WPB}}(\mu_Q)$. Note that since $\sigma_P = 0$ then the KL-divergence is undefined, while the $W_1$ distance equals exactly $\|\mu_Q - \mu_P\|_2$.

Figures 2a, 2b and 2c show the corresponding plots. The 'Training loss' column shows the empirical risk (57), i.e., the averaged loss of the learned posterior on the training data. The 'Test loss' column shows the average loss of the learned posterior on a separate 'test' set of 10000 samples drawn from $\mathcal{D}$. In all the evaluated bounds, we use the confidence parameter $\delta = 0.05$. The 'UC bound' shows

Table 1: Linear regression experiment with $\sigma_{\mathbf{P}} = \mathbf{10^{-2}}, \sigma_{\mathbf{Q}} = \mathbf{10^{-3}}$. Each cell shows the mean over 10 independent runs and the 95% confidence interval in parenthesis.

| # samples | Train risk | Test risk | UC bound | WPB bound | KLPB bound |
|---|---|---|---|---|---|
| 100 | 0.0211 (0.0010) | 0.0208 (0.0001) | 6.6176 (0.0010) | 2.2652 (0.0010) | 0.3861 (0.0010) |
| 200 | 0.0206 (0.0009) | 0.0208 (0.0001) | 4.6850 (0.0009) | 1.6080 (0.0009) | 0.2814 (0.0009) |
| 300 | 0.0214 (0.0006) | 0.0209 (0.0001) | 3.8298 (0.0006) | 1.3177 (0.0006) | 0.2357 (0.0006) |
| 400 | 0.0205 (0.0005) | 0.0208 (0.0001) | 3.3187 (0.0005) | 1.1433 (0.0005) | 0.2070 (0.0005) |

Table 2: Linear regression experiment with $\sigma_{\mathbf{P}} = \mathbf{10^{-4}}, \sigma_{\mathbf{Q}} = \mathbf{10^{-3}}$. Each cell shows the mean over 10 independent runs and the 95% confidence interval in parenthesis.

| # samples | Train risk | Test risk | UC bound | WPB bound | KLPB bound |
|---|---|---|---|---|---|
| 100 | 0.0211 (0.0010) | 0.0208 (0.0001) | 6.6176 (0.0010) | 0.7569 (0.0010) | 1.5787 (0.0010) |
| 200 | 0.0206 (0.0009) | 0.0208 (0.0001) | 4.6850 (0.0009) | 0.5424 (0.0009) | 1.1199 (0.0009) |
| 300 | 0.0214 (0.0006) | 0.0209 (0.0001) | 3.8298 (0.0006) | 0.4482 (0.0006) | 0.9186 (0.0006) |
| 400 | 0.0205 (0.0005) | 0.0208 (0.0001) | 3.3187 (0.0005) | 0.3906 (0.0005) | 0.7974 (0.0005) |

the sum of the empirical risk and the UC generalization gap bound (49). The 'WPB bound' is the Wasserstein-PB bound evaluated by equation (58), and the 'KLPB bound' is evaluated by equation (59). The results clearly show the improved tightness of the WPB bound over the UC bound, for the two choices of a prior distribution. The KLPB bound, also shows relatively tight values, as expected from an algorithm- and data-dependent bound. However, for the narrower prior distribution ($\sigma_P = 10^{-4}$), the KLPB bound is significantly looser than the WPB bound. That is expected from the properties of the KL-divergence, which can tend to $\infty$ if $\sigma_P \to 0$, as opposed to the Wasserstein distance. In the extreme case of $\sigma_P = 0$ the KLPB bound is undefined, while the WPB exhibits a considerable improvement over the UC bound. The results confirm that the WPB generally improves over UC bounds, and may be tighter than the KLPB bound, depending on the prior and posterior distributions.

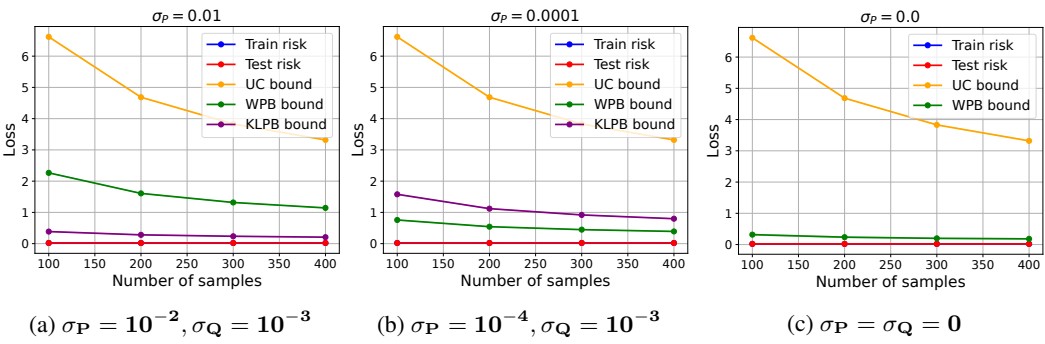

(a) $\sigma_{\mathbf{P}} = 10^{-2}, \sigma_{\mathbf{Q}} = 10^{-3}$     (b) $\sigma_{\mathbf{P}} = 10^{-4}, \sigma_{\mathbf{Q}} = 10^{-3}$     (c) $\sigma_{\mathbf{P}} = \sigma_{\mathbf{Q}} = 0$

Figure 2: Linear regression experiment. Note that the the 95% confidence interval is too small to be discernible in the plots, and that blue train risk plot is not visible since it is very close to the test risk.

# D  Appendix: Technical Lemmas

**Lemma 18.** *Let $A \subset \mathbb{R}$ be bounded and non-empty, and let $f : (0, \infty) \to \mathbb{R}$ be continuous and monotone non-decreasing. Then $f(\sup A) = \sup f(A)$, where we defined $\sup A \stackrel{def}{=} \sup_{a \in A} a$, and $\sup f(A) \stackrel{def}{=} \sup_{a \in A} f(a)$; that is, $f(A)$ is the image of the set $A$ under $f$.*

*Proof.* First notice that $f(\sup A) \geq \sup f(A)$ by monotonicity, because $a \leq \sup A$ for all $a \in A$.

For the other inequality, $f(\sup A) \leq \sup f(A)$ we need to also use continuity: let $\varepsilon > 0$; by continuity there exists $\delta > 0$ such that for every $a \in A$ such that $a \geq \sup A - \delta$ it holds that

Table 3: Linear regression experiment with $\sigma_{\mathbf{P}} = \mathbf{0}, \sigma_{\mathbf{Q}} = \mathbf{0}$. Each cell shows the mean over 10 independent runs and the 95% confidence interval in parenthesis.

| # samples | Train risk | Test risk | UC bound | WPB bound | KLPB bound |
|---|---|---|---|---|---|
| 100 | 0.0211 (0.0010) | 0.0208 (0.0001) | 6.6176 (0.0010) | 0.3175 (0.0177) | undefined |
| 200 | 0.0206 (0.0009) | 0.0208 (0.0001) | 4.6850 (0.0009) | 0.2363 (0.0136) | undefined |
| 300 | 0.0214 (0.0006) | 0.0209 (0.0001) | 3.8298 (0.0006) | 0.1989 (0.0087) | undefined |
| 400 | 0.0205 (0.0005) | 0.0208 (0.0001) | 3.3187 (0.0005) | 0.1824 (0.0127) | undefined |

$f(a) \geq f(\sup A) - \varepsilon$ (there exists such $a$ by the definition of the supremum). By monotonicity this implies that $\sup f(A) \geq f(\sup A) - \varepsilon$. Since the latter inequality holds for every $\varepsilon > 0$, we conclude that $\sup f(A) \geq f(\sup A)$ as required. $\qquad\square$

**Lemma 19** (Wasserstein distance between truncated Gaussian distributions). *Let $X^{(1)}$ and $X^{(2)}$ be the Gaussian random vectors in $\mathbb{R}^d$, with distributions $\mathcal{N}(\mu_1, \Sigma_1)$, and $\mathcal{N}(\mu_1, \Sigma_2)$ respectively. Let $P_{\mathbb{B}_r} : \mathbb{R}^d \to \mathbb{B}_r$ be a projection operator onto the an $r$-radius ball around the origin, $P_{\mathbb{B}_r}(x) \overset{def}{=}$ $\operatorname{argmin}_{x' \in \mathbb{B}_r} \|x - x'\|_2$, where $\mathbb{B}_r = \{x \in \mathbb{R}^d : \|x\|_2 \leq r\}$. Assume that $r \geq \sqrt{\|\mu_j\|_2^2 + \|\Sigma_j\|_F^2}$ for $j = 1, 2$. Denote the distribution measures of $X^{(1)}$ and $X^{(2)}$ as $\nu_1$ and $\nu_2$ respectively. Let $P_{\mathbb{B}_r}\sharp\nu_1$ and $P_{\mathbb{B}_r}\sharp\nu_2$ be the push-forward measures of $\nu_1$ and $\nu_2$, respectively, under the operator $P_{\mathbb{B}_r}$. Then*

$$W_1(P_{\mathbb{B}_r}\sharp\nu_1, P_{\mathbb{B}_r}\sharp\nu_2) \leq W_1(\nu_1, \nu_2) + \sum_{j=1}^{2} \sqrt{\frac{\pi}{2} \|\Sigma_j\|_{2,2}} \, \operatorname{erfc}\left( \frac{r - \sqrt{\|\mu_j\|_2^2 + \|\Sigma_j\|_F^2}}{\sqrt{2 \|\Sigma_j\|_{2,2}}} \right),$$

*where $W_1(Q, P)$ denotes the $1^{st}$ order Wasserstein distance with the $L_2$ metric.*

*Proof.* Using the triangle inequality of Wasserstein distances (Clement & Desch, 2008; Thorpe, 2018) twice we get

$$W_1(P_{\mathbb{B}_r}\sharp\nu_1, P_{\mathbb{B}_r}\sharp\nu_2) \leq W_1(P_{\mathbb{B}_r}\sharp\nu_1, \nu_1) + W_1(\nu_1, \nu_2) + W_1(\nu_2, P_{\mathbb{B}_r}\sharp\nu_2). \tag{60}$$

Notice that

$$W_1(\nu_2, P_{\mathbb{B}_r}\sharp\nu_2) = \inf_{\gamma \in \Gamma(\nu_2, P_{\mathbb{B}_r}\sharp\nu_2)} \int_{\mathbb{R}^d \times \mathbb{R}^d} \|x - x'\|_2 \, d\gamma(x, x') \tag{61}$$

$$\leq \int_{\mathbb{R}^d} \|x - P_{\mathbb{B}_r}(x)\|_2 \, d\nu_2(x)$$

$$\overset{(i)}{=} \int_{\mathbb{R}^d} [\|x\|_2 - r]_+ d\nu_2(x)$$

$$= \mathbb{E}\left\{ [\|X_2\|_2 - r]_+ \right\}$$

$$\overset{(ii)}{=} \mathbb{E}\left\{ [V - r]_+ \right\}$$

$$\overset{(iii)}{=} \int_{t=0}^{\infty} \mathbb{P}\left\{ [V - r]_+ > t \right\} dt$$

$$\overset{(iv)}{=} \int_{t=0}^{\infty} \mathbb{P}\{V > r + t\} dt,$$

where in *(i)* we used the notation $[t]_+ \overset{def}{=} \begin{cases} t & t > 0 \\ 0 & t \leq 0 \end{cases}$ and the equality holds since $P_{\mathbb{B}_r}(x) = x$ for $\|x\| \leq r$, and $\|P_{\mathbb{B}_r}(x) - x\| = \|x\|_2 - r$ for $\|x\| > r$ (by the properties of the projection onto the a $L_2$ ball). In *(ii)* we use the definition of the random variable $V \overset{def}{=} \|X^{(2)}\|_2$. In *(iii)* we used the tail sum formula for the expectation. Equality *(iv)* holds since the corresponding events are equivalent.

Let $Z$ be a standard Gaussian random vector in $\mathbb{R}^d$, i.e., $Z \sim \mathcal{N}(\bar{\mathbf{0}}, I)$.

By Wainwright (2019), Thm. 2.26 (one-sided variant) we have that

$$\mathbb{P}\{f(Z) - \mathbb{E}[f(Z)] \geq s\} \leq \exp\left(-\frac{s^2}{2L^2}\right),$$

for any $s \geq 0$ and for any function $f : \mathbb{R}^d \to \mathbb{R}$ that is $L$-Lipschitz w.r.t. the $L_2$ metric.

In particular, for the function $f(z) \overset{\text{def}}{=} \left\|\mu_2 + \Sigma_2^{1/2} z\right\|_2$ we have for any $s \geq 0$

$$\mathbb{P}\left\{\left\|\mu_2 + \Sigma_2^{1/2} Z\right\|_2 - \mathbb{E}\left[\left\|\mu_2 + \Sigma_2^{1/2} Z\right\|_2\right] \geq s\right\} \leq \exp\left(-\frac{s^2}{2\left\|\Sigma_2^{1/2}\right\|_{2,2}^2}\right) \tag{62}$$

$$\overset{(i)}{=} \exp\left(-\frac{s^2}{2\left\|\Sigma_2\right\|_{2,2}}\right),$$

since $f$ is Lipschitz with constant $\left\|\Sigma_2^{1/2}\right\|_{2,2}$ where $\|\cdot\|_{2,2}$ is the operator norm defined by $\|A\|_{2,2} = \sup_{\|x\|_2=1} \|Ax\|_2$ for any $A \in \mathbb{R}^{d \times d}$. Equality *(i)* holds since

$$\left\|\Sigma_2^{1/2}\right\|_{2,2}^2 = \left(\sup_{\|x\|_2=1} \left\|\Sigma_2^{1/2} x\right\|_2\right)^2 = \sup_{\|x\|_2=1} \left\|\Sigma_2^{1/2} x\right\|_2^2 = \sup_{\|x\|_2=1} x^\top \Sigma_2 x = \|\Sigma_2\|_{2,2}.$$

Notice that

$$\mathbb{E}\left[\left\|\mu_2 + \Sigma_2^{1/2} Z\right\|_2\right] = \mathbb{E}\left[\sqrt{\left\|\mu_2 + \Sigma_2^{1/2} Z\right\|_2^2}\right]$$

$$\overset{(i)}{\leq} \sqrt{\mathbb{E}\left\{\left\|\mu_2 + \Sigma_2^{1/2} Z\right\|_2^2\right\}}$$

$$= \sqrt{\mathbb{E}\left\{\left(\mu_2 + \Sigma_2^{1/2} Z\right)^\top \left(\mu_2 + \Sigma_2^{1/2} Z\right)\right\}}$$

$$\overset{(ii)}{=} \sqrt{\|\mu_2\|_2^2 + \|\Sigma_2\|_F^2},$$

where *(i)* is by Jensen's inequality, and in *(ii)* we used the fact that $Z \sim \mathcal{N}(\bar{0}, I)$ and $\|\cdot\|_F$ denotes the Frobenius Norm, defined by $\|A\|_F \overset{\text{def}}{=} \sqrt{\sum_{i,j} A_{i,j}^2}$.

Therefore, using (62), we get

$$\mathbb{P}\left\{\left\|\mu_2 + \Sigma_2^{1/2} Z\right\|_2 - \sqrt{\|\mu_2\|_2^2 + \|\Sigma_2\|_F^2} \geq s\right\} \leq \exp\left(-\frac{s^2}{2\left\|\Sigma_2\right\|_{2,2}}\right).$$

Note that the random variable $V \overset{\text{def}}{=} \left\|X^{(2)}\right\|_2$ is equal, in distribution, to the random variable $\left\|\mu_2 + \Sigma_2^{1/2} Z\right\|_2$, and therefore we also have for $s \geq 0$ that

$$\mathbb{P}\left\{V - \sqrt{\|\mu_2\|_2^2 + \|\Sigma_2\|_F^2} \geq s\right\} \leq \exp\left(-\frac{s^2}{2\left\|\Sigma_2\right\|_{2,2}}\right).$$

For any $t \geq 0$, set $s := t + r - \sqrt{\|\mu_2\|_2^2 + \|\Sigma_2\|_F^2}$. Since we assume that $r \geq \sqrt{\|\mu_2\|_2^2 + \|\Sigma_2\|_F^2}$, we have that $s \geq 0$. Therefore we have

$$\mathbb{P}\{V > r + t\} \leq \exp\left(-\frac{\left(t + r - \sqrt{\|\mu_2\|_2^2 + \|\Sigma_2\|_F^2}\right)^2}{2\left\|\Sigma_2\right\|_{2,2}}\right).$$

Hence, by (61) we have

$$W_1(\nu_2, P_{\mathbb{B}_r}\sharp\nu_2) \leq \int_{t=0}^{\infty} \mathbb{P}\{V > r + t\}\mathrm{d}t$$

$$\leq \int_{t=0}^{\infty} \exp\left(-\frac{\left(t + r - \sqrt{\|\mu_2\|_2^2 + \|\Sigma_2\|_F^2}\right)^2}{2\|\Sigma_2\|_{2,2}}\right)\mathrm{d}t$$

$$= \sqrt{\frac{\pi}{2}\|\Sigma_2\|_{2,2}}\,\mathrm{erfc}\left(\frac{r - \sqrt{\|\mu_2\|_2^2 + \|\Sigma_2\|_F^2}}{\sqrt{2\|\Sigma_2\|_{2,2}}}\right).$$

By symmetry we have a similar bound for $W_1(P_{\mathbb{B}_r}\sharp\nu_1, \nu_1)$. To conclude, using (60) we get

$$W_1(P_{\mathbb{B}_r}\sharp\nu_1, P_{\mathbb{B}_r}\sharp\nu_2) \leq W_1(P_{\mathbb{B}_r}\sharp\nu_1, \nu_1) + W_1(\nu_1, \nu_2) + W_1(\nu_2, P_{\mathbb{B}_r}\sharp\nu_2)$$

$$\leq W_1(\nu_1, \nu_2) + \sum_{j=1}^{2}\sqrt{\frac{\pi}{2}\|\Sigma_j\|_{2,2}}\,\mathrm{erfc}\left(\frac{r - \sqrt{\|\mu_j\|_2^2 + \|\Sigma_j\|_F^2}}{\sqrt{2\|\Sigma_j\|_{2,2}}}\right).$$

$\square$