# OpenReview forum: "Integral Probability Metrics PAC-Bayes Bounds"
_NeurIPS.cc/2022/Conference — NeurIPS 2022 Accept_

### Official Review · Reviewer_TFE4 · 2022-06-24

**Rating:** 6
**Confidence:** 4
**Soundness:** 3 good
**Presentation:** 2 fair
**Contribution:** 3 good

**Summary:**

Authors open the classical PAC-Bayesian framework to new type of divergences, going beyond the classical KL-divergence or the $f$-divergences. They introduce the general notion of Integral Probability Metric (IPM) and they derived from it general templates for a new type of IPM-PAC-Bayesian bounds.

From those general bounds, authors provide new concrete results for two classical divergences: the TV distance and the Wasserstein distance.

**Questions:**

Do authors have an argument to counter my claim on Prop. 4 's veracity?

- Is there any cases where one can compute TV-distance nor the Wasserstein one? To the local nature (see the major issue of Strengths&Weaknesses section) of all the proposed result I believe insisting on the practical extensions is now essential on this work.

- I am wondering about the choice of the universal constant in Cor. 8. Is it an accessible quantity? I am wondering if the proposed bound remains fully empirical.

- In Th. 11, do authors have an instance of a non-Lipschitz function verifying the assumption of being Lipschitz with high probability?


**Limitations:**

Autors are clear on their framework and the remits of their contributions.



**Strengths And Weaknesses:**

MAJOR ISSUE:

To my understanding (please correct me if there is an argument that I missed), the preliminary theorem (Prop 4.) is not correct. Authors stated this result holds "with probability $1-\delta$, for all $Q\in\mathcal{M}(\mathcal{H})$..." while the correct formulation is  "for all $Q\in\mathcal{M}(\mathcal{H})$,with probability $1-\delta$..."

I provide below a corrected version of author's proof which starts from the line 458 of the appendix.

NEW PROOF--

Inequalities (23) and (24) imply $\\forall (Q,P)\\in \\mathcal{M}(\\mathcal{H})^2,$

 $\\mathbb{E}\_{S \\sim \\mathcal{D}^{m}} \\exp \\left(\\underset{h \\sim Q}{\\mathbb{E}}\\left[f\_{S}(h)\\right]-\\gamma_{\\mathcal{F}_{S}}(Q, P)\\right) \\leq m .$

Therefore, by Markov's inequality, for any $t>0$ we have  $\forall (Q,P)\in \mathcal{M}(\mathcal{H})^2,$

\begin{multline*}
  \mathbb{P}\_{S \sim \mathcal{D}^{m}}\left(\exp \left(\underset{h \sim Q}{\mathbb{E}}\left[f_{S}(h)\right]-\gamma_{\mathcal{F}\_{S}}(Q, P)\right) \geq t\right) \\
   \leq \frac{\mathbb{E}\_{S \sim \mathcal{D}^{m}} \exp \left(\underset{h \sim Q}{\mathbb{E}}\left[f_{S}(h)\right]-\gamma_{\mathcal{F}\_{S}}(Q, P)\right)}{t} \leq \frac{m}{t} .
\end{multline*}


Or, equivalently, $\forall (Q,P)\in \mathcal{M}(\mathcal{H})^2,$
$$
\mathbb{P}\_{S \sim \mathcal{D}^{m}}\left(\underset{h \sim Q}{\mathbb{E}}\left[f\_{S}(h)\right]-\gamma_{\mathcal{F}\_{S}}(Q, P) \geq \ln (t)\right) \leq \frac{m}{t}
$$

Let $\delta \in(0,1)$, we set $t=\frac{m}{\delta}$, and plug in $f_{S}(h) \stackrel{\text { def }}{=} 2(m-1) \Delta_{S}^{2}(h)$ to get
$$
\forall (Q,P)\in \mathcal{M}(\mathcal{H})^2,\mathbb{P}\_{S \sim \mathcal{D}^{m}}\left(\underset{h \sim Q}{\mathbb{E}}\left(2(m-1) \Delta\_{S}^{2}(h)\right)-\gamma_{\mathcal{F}\_{S}}(Q, P) \geq \ln (m / \delta)\right) \leq \delta
$$
Rearranging the above, we have that, for all $Q,P \in \mathcal{M}(\mathcal{H})^2$, with probability at least $1-\delta$ over the samples $S \sim \mathcal{D}^{m}$, the following inequality holds
$$
\underset{h \sim Q}{\mathbb{E}}\left(\Delta\_{S}^{2}(h)\right)<\frac{\gamma_{\mathcal{F}\_{S}}(Q, P)+\ln (m / \delta)}{2(m-1)} .
$$
By Jensen's inequality applied to $\Delta\_{S}(h)^{2}$
$$
\left(\underset{h \sim Q}{\mathbb{E}} \Delta_{S}(h)\right)^{2} \leq \underset{h \sim Q}{\mathbb{E}}\left(\Delta_{S}^{2}(h)\right) \leq \frac{\gamma_{\mathcal{F}}(Q, P)+\ln (m / \delta)}{2(m-1)}
$$
The theorem follows for any pair $(Q,P)$ with probability $1-\delta$ by taking the square root of both sides.

--

The problem with author's proof is that they did not realise that they were applying Markov's inequality on a random variable depending on the sample $S$ and the distributions $Q,P$. **Modifying the posterior distribution changes the variable onto which we apply Markov's inequality and then requires a novel application of this bound.** That is why Prop 4. cannot hold with high probability for all posteriors (except on the case of a finite posterior class $C$ where one can apply an union bound on the events and maintain the result at the cost of an extra $\log(|C|)$).

That is why the change of measure inequality is so important in classical PAC-Bayesian proofs: it allows a deterministic control over the whole class of posterior by bounding a moment depending on $Q$ by a KL divergence plus a random variable depending only on the prior $P$ and the sample $S$. Then one can apply a single time Markov's inequality on this random variable to obtain an uniform PAC bound over all the posteriors.


Thus, this apparently small modification of the result has actually huge consequences:

- All the theorems provided in the paper lose the strength of having a global control over the posterior class : they only have a local control over a single posterior, which is not catastrophic but I believe this fact changes the spirit of the paper. Indeed, I think authors should now also focus on constructive procedure to exhibit concrete posteriors (which still benefits from author's theoretical guarantees)

- The place of the change of measure inequality is crucial to obtain strong global guarantees and is not only a mathematical technique to involve KL-divergence. Authors should also check and refer to the work of (Picard-Weibel and Guedj, 2022) which tries to generalise the change of measure inequality to others type of divergences.


Other points:

- This major point being said, I am not sure on how I should consider the rest of the new resuts of the paper as they are all derived from Prop. 4.  I see plenty of results which are now local on the choice of the couple $(Q,P)$. Then I am wondering if one can derive some concrete optimisation routes from those upper bounds if one restricts to specific classes. More generally, I did not see much discussion about the range of the proposed results which is a shame, especially when authors still have more than a half page available.

- I would have appreciated more discussion about the proposed theorems, their novelty and the interest of considering such new distances (although I admit  the end of Sec. 5 goes in this direction) given the extra space authors still have

- I find interesting the idea of exploit IPM onto a PAC-Bayes framework. It may be a fruitful tool to extract PAC-Bayes from KL and $f$-divergences.


CONCLUSION: My main problem with this work is Prop. 4's proof. To me, this point weakens the paper a lot as it shows that for instance the TV distance nor the Wasserstein one are 'less adapted' to PAC-Bayesian than the KL divergence as it allows to only recovers local results for a similar framework. This point, to me, should be at least acknowledged and a discussion about the change of measure inequality should be proposed in this work. However, I think that with a more careful study, this line of research could be fruitful and I do not contest the novelty of the proposed approach. A convincing axis of work would be to focus on the practical extensions of their work and exploit the fact that, according to the authors, Wasserstein distance is potentially more stable in practice. Having concrete extensions would for me, compensate the local behavior of those results.

---

> ### Author Response · Authors · 2022-08-02
> **Response to Reviewer TFE4**
>
>
>  We thank the reviewer for carefully inspecting the proof and for offering a correction; this is truly generous and collegial.
> Fortunately, the gap in the proof of Proposition 4 can be fixed by a couple of lines;
> in fact, we noticed this just after the submission deadline.
>
> The corrected proof appears in the current version of the submission. For convenience, we also include the modified parts of the proof here, with the changes highlighted in red.
>
> $\color{red} \text{Taking the supremum over } Q \in \mathcal{M}(\mathcal{H})$, and an expectation over samples $S \sim \mathcal{D}^m$, we have for any  $P \in \mathcal{M}(\mathcal{H})$,
> \begin{align}
> E_{S \sim \mathcal{D}^m} {\color{red} \sup_{Q}} \exp (E_{h \sim Q} [f_{S}(h)]-\gamma_{F_S}(Q, P)) & \le E_{S \sim \mathcal{D}^m} {\color{red} \sup_{Q}} (E_{h\sim P} \exp[{f_{S}(h)] } )\nonumber \quad (23) \\\\
>  & =E_{S \sim \mathcal{D}^m}  E_{h\sim P}  \exp[{f_{S}(h)]} \nonumber  \\\\
>  & = E_{h\sim P}E_{S \sim \mathcal{D}^m}\exp[f_{S}(h)], \nonumber
> \end{align}
> where the last equality is obtained by the prior's independence from the sample, and from Fubini's theorem.
>
> Inequalities (23) and (24) imply
> \begin{align}
> E_{S \sim \mathcal{D}^m}  {\color{red} \sup_{Q}} \exp ( E_{h \sim Q}[ f_{S}(h)] -\gamma_{F_S}(Q, P))  & \le m. \nonumber
> \end{align}
> Therefore, by Markov's inequality, for any $t >0$ we have
> \begin{align}
> P_{S \sim \mathcal{D}^m}\left( {\color{red} \sup_{Q}} [\exp (E_{h \sim Q}[f_{S}(h)] - \gamma_{F_S}(Q, P))] \geq t\right) & \le\frac{m}{t}. \nonumber
> \end{align}
> Or, equivalently,
> \begin{align}
> P_{S \sim \mathcal{D}^m}\left( {\color{red} \sup_{Q}} [E_{h \sim Q}[f_{S}(h)] - \gamma_{F_S}(Q,P)] \ge \ln(t) \right) & \le\frac{m}{t}. \nonumber
> \end{align}
>
> Let $\delta \in (0,1)$, we set $t= \frac{m}{\delta}$, and plug in $f_{S}(h) :=2(m-1) \Delta_{S}^{2}(h)$ to get
> \begin{align}
> P_{S\sim\mathcal{D}^m}\left({\color{red}\sup_{Q}}[E_{h \sim Q}\left(2(m-1)\Delta_{S}^{2}(h)\right) - \gamma_{F_S}(Q, P) ] \ge \ln (\frac{m}{\delta}) \right) & \le \delta. \nonumber
> \end{align}
> Therefore, the complementary event satisfies
> \begin{align}
> P_{S \sim \mathcal{D}^m}\left(  {\color{red} \sup_{Q}} [E_{h \sim Q}\left(2(m-1)\Delta_{S}^{2}(h)\right) - \gamma_{F_S}(Q, P)]< \ln (\frac{m}{\delta}) \right) & \ge 1 -\delta. \nonumber
> \end{align}
> Thus, for any  $P \in \mathcal{M}(\mathcal{H})$, with a probability of a least
> $1-\delta$ over the samples $S \sim \mathcal{D}^m$, the following inequality holds for all $Q \in \mathcal{M}(\mathcal{H})$
> \begin{align}
> E_{h \sim Q}\left(\Delta_S^{2}(h)\right) & < \frac{\gamma_{F_S}(Q, P) + \ln(\frac{m}{\delta})}{2(m-1)}. \nonumber
> \end{align}
>
> This correction settles the major issue raised by the reviewer and demonstrates that our results hold with high probability over all posterior distributions, as was claimed.
> Based on this correction, we will appreciate it if the reviewer reconsiders the score.
>
> **Other Issues**
>
> -  We thank you for the referral to (Picard-Weibel and Guedj, 2022). We will take a close look and discuss it in future revisions.
>
> - "More generally, I did not see much discussion about the range of the proposed results". What do you mean by "range of the results"?
>
> **Question Answers**
>
> 1.   For Gaussian distributions, the Wasserstein metric has an analytical form (but the TV distance does not have an analytical form for interesting distributions, as far as we know). We will discuss this important point in future revisions.
>
> 2. Regarding the universal constant in Cor. 8 - this type of bound originates from (Talagrand 1994) and
> as far as we know, there is no explicit value of the constant in the literature.
> Obtaining the constant involves careful computations of covering numbers and using the chaining method (e.g., based on Theorems 1.16 and 1.17 in Lugosi 2002).
> Since our focus was not on the numerical evaluation of the bounds, we did not include this in our work. In future revisions, we will add a note regarding this. However, there are other VC-type bounds with explicit constants (e.g. a value of $4$), but with an extra $\log(m)$ factor (e.g., Vapnik 2000, Sect 3.4). we will add a note regarding this.
>
> 3. The generalization gap is a random function (as it is sample-dependent), we proved it is Lip w.h.p. for the examples of Corollary 15 (linear regression) and Theorem 12 (finite class with Lip loss). In both cases, a `bad' sample can result in non Lip function (e.g. step function), but in high probability, Lip does hold.
>
> **References**
>
> - M. Talagrand, Sharper Bounds for Gaussian and Empirical Processes, The Annals of Probability 1994
>
> -  G. Lugosi, Pattern classification and learning theory, Springer 2002
>
> - V. Vapnik, The nature of statistical learning theory, Springer 2000
>
> - S. Shalev-Shwartz and S. Ben-David, Understanding Machine Learning: From Theory, Cambridge university press 2014

---

> > ### Comment · Reviewer_TFE4 · 2022-08-03
> > **I still have one question about Prop.4's proof**
> >
> > I thank the authors for their response.
> >
> > First of all I thank the authors for the correction, unfortunately it seems to me that one line of the proof still not holds. I am putting the details below:
> >
> > **My vision of your proof**
> >
> > Starting from this (correct) line:
> >
> > Therefore, by Markov's inequality, for any $t>0$ we have
> >
> > $ \mathbb{P}\_{S \sim \mathcal{D}^{m}}\left( sup_{Q}\exp \left(\underset{h \sim Q}{\mathbb{E}}\left[f_{S}(h)\right]-\gamma_{\mathcal{F}\_{S}}(Q, P)\right) \geq t\right)  \leq \frac{m}{t} $
> >
> > Or, equivalently:
> >
> > $  \mathbb{P}\_{S \sim \mathcal{D}^{m}}\left( \ln\left(sup_{Q}\exp \left(\underset{h \sim Q}{\mathbb{E}}\left[f_{S}(h)\right]-\gamma_{\mathcal{F}\_{S}}(Q, P)\right)\right) \geq \ln(t)\right) \leq \frac{m}{t} $
> >
> > To me it seems that authors implictly affirmed that :
> >
> > $ \ln\left(sup_{Q}\exp \left(\underset{h \sim Q}{\mathbb{E}}\left[f_{S}(h)\right]-\gamma_{\mathcal{F}\_{S}}(Q, P)\right)\right) =  sup_{Q}\ln\left(\exp \left(\underset{h \sim Q}{\mathbb{E}}\left[f_{S}(h)\right]-\gamma_{\mathcal{F}\_{S}}(Q, P)\right) \right)  $
> >
> > $= sup_Q \underset{h \sim Q}{\mathbb{E}}\left[f_{S}(h)\right]-\gamma_{\mathcal{F}\_{S}}(Q, P) $
> >
> > which is not obvious. Indeed, because $\ln$ is an increasing function, we immediately have :
> >
> > $ \ln\left(sup_{Q}\exp \left(\underset{h \sim Q}{\mathbb{E}}\left[f_{S}(h)\right]-\gamma_{\mathcal{F}\_{S}}(Q, P)\right)\right) \geq sup_{Q}\ln\left(\exp \left(\underset{h \sim Q}{\mathbb{E}}\left[f_{S}(h)\right]-\gamma_{\mathcal{F}\_{S}}(Q, P)\right) \right)  $
> >
> > but the other inequality is far from obvious to me.
> >
> > Again, is there a point that I missed?
> >
> >
> > **Other Issues**
> >
> > - What I meant by "the range of the results" is some contextual discussion about your results, for instance in which context do we use the loss-gradient UC property ? Is there some practical extensions where the Wasserstein distance is used in pratice?
> >
> > **Questions**
> >
> > 1. Thank you for your answer, I think this point should appear in the main document.
> >
> > 2. Thank you for the clarification, to me it is important to precise that explicit constants can appear as having fully empirical bound is a crucial step to transform your upper bound onto a learning objective I think this should be added as a supplementary corollary in order to make a supplementary step toward PAC-Bayes practitioners.
> >
> > 3 Thank you for your instance.

---

> > > ### Author Response · Authors · 2022-08-05
> > > **Response to the question from reviewer TFE4**
> > >
> > > We thank the reviewer again for carefully inspecting the proof.
> > > The transition holds because $\ln(x)$ is continuous and monotone on $(0,\infty)$.
> > > In more detail, it follows from the next claim.
> > >
> > > **Claim**. Let $A\subset \mathbb{R}$ be bounded and non-empty, and let $f:(0,\infty) \to \mathbb{R}$ be continuous and monotone non-decreasing.
> > > Then $f(\sup A) = \sup f(A)$, where we defined $\sup A := \sup_{a \in A}a$, and $\sup f(A) := \sup_{a\in A} f(a)$; that is, $f(A)$ is the image of the set $A$ under $f$.
> > >
> > > In our context the function $f$ is $\ln(x)$ and the set
> > > $A$ is the set of numbers $[\exp( \mathbb{E}_Q[f_S(h)] - \gamma(Q,P) ) : Q \text{ is a distribution}]$.
> > > We hope this clarifies the step and we will elaborate on it in the next version.
> > >
> > > **Proof of Claim**.
> > > First notice that $f(\sup A) \geq \sup f(A)$ by monotonicity, because $a \leq \sup A$ for all $a \in A$.
> > >
> > > For the other inequality, $f(\sup A) \leq \sup f(A)$ we need to also use continuity:
> > > let $\varepsilon > 0$; by continuity there exists $\delta>0$ such that
> > > for every $a \in A$ such that $a \ge \sup A - \delta$ it holds that $f(a) \geq f(\sup A) - \varepsilon$ (there exists such $a$ by the definition of the supremum).
> > > By monotonicity this implies that $\sup f(A) \geq f(\sup A) - \varepsilon$.
> > > Since the latter inequality holds for every $\varepsilon > 0$,
> > > we conclude that $\sup f(A) \geq f(\sup A)$ as required.
> > >
> > > Regarding "other issues", Corollary 15 shows that for a linear regression example where the loss-gradient UC property holds.
> > > Regarding contextual discussion and practical use of our work, we refer the reviewer to our general response.

---

> > > > ### Comment · Reviewer_TFE4 · 2022-08-05
> > > > **Final response**
> > > >
> > > > I thank the reviewer for their time.
> > > >
> > > > To me, there is no major point left motivating the reject of the paper, I then move my score to 6.

---

### Official Review · Reviewer_3Pda · 2022-07-09

**Rating:** 6
**Confidence:** 4
**Soundness:** 3 good
**Presentation:** 3 good
**Contribution:** 2 fair

**Summary:**

The paper presents new generalizations bounds that aesthetically look like a mix between uniform convergence and PAC-Bayesian bounds.   In one instantiation, the new bound can improve on uniform convergence bounds by a factor proportional to the square root of the TV distance between a posterior and prior distribution. The authors achieve their results thanks to a new IPM-based generalization guarantee.

**Questions:**

Have you done/can you include any experiments to compare the new bounds to previous PAC-Bayesians ones?

**Limitations:**

Some suggestions for improvement (mainly typos):
- It would be good to include an explicit comparison of the new bounds to those of, e.g., Neu and Lugosi (2022). (If space is an issue, at least include the bound of the latter in the appendix).
- Eq (4), f_S is not defined within Def 3
- Line 199: with the the zero-one -> with the zero-one
- Line 234: $K(m,\delta')$-Lipschitz? Do you mean just $K$-Lipschitzness with $K$ that may depend on $m$ and $\delta'$? If so, this is a little confusing---just say $K$-Lipschitz.
- Line 239-240: "is K-Lipschitz with high probability, where the rate of $K$ is preferably $O(1/m)$" Not clear what you mean by this. Clarify.
- Line 273: "Assume that for any fixed $z ∈ Z$, $f(h,z)$ is continuously differentiable for any $h \in H$". This is redundant as it is already a condition in Def 13.


**Strengths And Weaknesses:**

The new IPM-based generalization guarantees seem new and would be of interest to the ML community in my opinion. One desirable feature of these bounds is that they allow for degenerate posteriors (the posterior need not be absolutely continuous w.r.t. the prior), which is not allowed in standard PAC-Bayesian bounds.

In terms of weaknesses, the fact that uniform convergence is required in most bounds is not ideal. Especially, since the new bounds depend on the uniform convergence error which can be large for Neural Networks, for example. After all, having non-vacuous bounds for large classes was one of the main motivations behind the paper.

---

> ### Author Response · Authors · 2022-08-02
> **Response to Reviewer 3Pda**
>
> We thank the reviewer for the helpful feedback.
>
> The reviewer's concerns were very similar to those of reviewer 2X57, and are addressed in our general response
> above.
>
> We will expand the discussion related to Neu and Lugosi (2022) in a future revision. We emphasize, though, that their result holds in expectation, as opposed to our high probability results.
> Thank you for the rest of the suggestions. We included them in the revised version.

---

### Official Review · Reviewer_2x57 · 2022-07-11

**Rating:** 7
**Confidence:** 4
**Soundness:** 4 excellent
**Presentation:** 3 good
**Contribution:** 3 good

**Summary:**

The paper derives PAC-Bayes generalization bounds taking as complexity term integral probability metrics, in particular Total Variation and Wasserstein distance.
Existing bounds based on such complexities stand in expectation over samples from the data distribution, while the new results stand with high probability.
The principal requirements for deriving such bounds is that the loss is bounded and Uniform Convergence stands. The new certificates are potentially tighter than this type of classical results as they are algorithm and data dependent.

**Questions:**

My principal question is detailed in weaknesses 2. In other words, in which settings the new results improve on both classical Uniform Convergence bounds and PAC-Bayesian bounds?

**Limitations:**

The limitations of the work are adequately addressed.

**Strengths And Weaknesses:**

### strengths
1. To the best of my knowledge the results are novel and non-trivial, as it is not straightforward to derive bounds with Integral Probability Metric within the PAC-Bayesian framework. In my opinion, the main advantage of the new certificates is that they drop the requirement of absolute continuity, hence the prior distribution can have null mass on bad predictors of the hypothesis class.

2. The paper is generally well presented and the proofs are clear and easy to follow.

### weaknesses
1. I find the argument that this work should improve existing generalization bounds for over-parameterized model weak. lt is not clear whether the new factors really tighten the bounds enough to make them non-vacuous (at the very least). On the contrary introducing VC or Radamacher complexity measures in PAC-Bayes bounds might end up make them looser, as these complexity measures do not correlated well with generalization (see e.g. [1]). On the other hand, dropping the constraint of absolute continuity potentially allows to derive certificates for dirac posteriors, hence to remove the need to derandomize the bound for deterministic models. However, the IPM might become too big for the available sample size. An empirical analysis or reporting few example settings would have helped showing in which contexts the new results have practical interest.

2. The literature on early works on algorithm dependent guarantees is not reported. Please consider citing at least the following seminal works: Uniform Stability [2], Algorithmic Robustness [3].

### minors
#### Writing
Paragraph 40-54: specify when expectations/probabiltities are over data or over predictors; which uniform convergence assumptions do you refer to?
Discussion: which works are referred to as "localized based approaches"?

#### Typos
1. line 48: probabily -> probability
2. line 51: it can -> can
3. line 243: Wasserstein-BP -> Wasserstein-PB
4. Discussion: divergences -> distances

[1] Jiang, Yiding, et al. "Fantastic generalization measures and where to find them." ICLR (2020).

[2] Bousquet, Olivier, and André Elisseeff. "Stability and generalization." The Journal of Machine Learning Research 2 (2002): 499-526.

[3] Xu, Huan, and Shie Mannor. "Robustness and generalization." Machine learning 86.3 (2012): 391-423.

---

> ### Author Response · Authors · 2022-08-02
> **Response to Reviewer 2x57**
>
> We thank the reviewer for the helpful feedback.
>
> We agree that utilizing  UC bounds in the over-parameterized regime may lead to very loose bounds.
> Thank you for the reference to (Jiang et al. 2020), it clearly demonstrates this point. We will highlight this in a future revision.
> Please see our general response to this issue above.
>
> Regarding the reviewer's question, we expanded the discussion of the linear regression example of Cor. 15, to emphasize that there are cases where the IPM-PB is tighter than both the UC and KL-PB bounds (when the KL can be arbitrarily large). In the final version, we will strengthen this claim with a numerical evaluation.
>
> We will improve the literature review on algorithm-dependent guarantees. Thank you for the suggestion.
>
> **Minors**
> - The paragraph discusses probabilities over data. We will clarify this.
> - We refer to localized Rademacher complexity bounds. We will add relevant references in future revisions.

---

> > ### Comment · Reviewer_2x57 · 2022-08-08
> > **confirm initial recommendation**
> >
> > I thank the authors for their response. I keep my initial recommendation of accepting the paper, also in light of the correction of the proof of Proposition 4.

---

### Official Review · Reviewer_cRSg · 2022-07-11

**Rating:** 7
**Confidence:** 4
**Soundness:** 4 excellent
**Presentation:** 4 excellent
**Contribution:** 3 good

**Summary:**

The present work develops PAC-Bayes like generalization bounds using integral probability metrics in place of divergences between a prior and the posterior. The bounds combine aspects of data independent ‘uniform convergence’ bound (based on empirical process theory), and data-dependent bounds (as is typical of PB bound, as well as margin based bounds, etc.) in a way that generalizes both since the present work’s bounds are smaller than one would expect from either alone. The present work specializes this to two particular IPM’s, total variation distance and Wasserstein distance. The focus is on theoretical results;  specific examples and numerical evaluation of bounds are not discussed.

**Questions:**

NA

**Limitations:**

Lack of experimental/numerical results showing the new bounds are materially better in practice than existing ones.

**Strengths And Weaknesses:**

The paper has a number of strengths, which lead me to support its acceptance.
The writing is very clear, easy to follow, concise and accurate. The paper was a pleasure to read.
The results are interesting, useful, and (to my knowledge) completely original.
The proofs are straight forward and easy to follow. Thus the results appear to be sound.
An aspect of the work which I find quite compelling, and an especially good fit for Neurips, is that the Wasserstein bounds do not require the posterior to be abs.continuous w.r.t. the prior, which means the bounds can be applied to SGD  training of general models, without modification to ensure abs.continuity. This is in contrast to prior work where similar bounds are expressed in terms of divergences.

The main weakness that I would like to point out, and the only item which addressed well would likely lead me to increase my score, is the lack of experimental or numerical evidence that shows that the theoretical improvements to standard bounds made herein are material in practical settings. An example of a model+training method where UC and PB bounds are both loose and the bounds of the present work are tight would materially improve the paper.

---

> ### Author Response · Authors · 2022-08-02
> **Response to Reviewer cRSg**
>
> We thank the reviewer for the helpful feedback.
>
>  Regarding numerical results - we agree that such results can be valuable. Please see our general response to this issue above.

---

### Author Response · Authors · 2022-08-02
**General response**


We thank the reviewers for taking the time to carefully read our work (remarkably, even the proofs!),
and for their thoughtful comments and suggestions to improve it.

The reviewers seem to agree that the proposed approach (of combining uniform convergence and algorithm-dependent bounds) is novel and non-trivial.
The reviewers also appreciate the fact that the resulting bounds circumvent some disadvantages of classical uniform convergence and PAC-Bayes bounds (e.g. that in contrast with traditional PAC-Bayes, our bound does not require absolute continuity.)

The main issue the reviewers raise is the lack of explicit numerical/experimental demonstration of our bounds. We completely agree that such results are most important: we have included in the current version a simple example of explicit bounds in the context of linear regression, facilitating a simple intuitive comparison between different bound types. We will include a numerical demonstration of this in the final version.
Nevertheless, we would like to stress that our main aim in this work is demonstrating our approach in a mathematical clean and didactic way;
in other words, we optimized for clarity and simplicity and tried to state and prove bounds which lay out the approach as clearly as we can.
We believe that in order to obtain non-vacuous bounds for **specific** learning algorithms and tasks of interest,
one will need to refine our general approach, e.g. by replacing uniform convergence with localized uniform convergence.
We view this as an important direction for future research and hope that our work will inspire and encourage the community to explore it.

Reviewer TFE4 pointed out a gap in our proof of Proposition 4.
We would like to thank the reviewer for taking the time to read the proof and for offering a correction (which yields a weaker version of the proposition).
This is truly generous and collegial!
Fortunately, the proof of the original proposition can be fixed by a couple of lines (in fact, we noticed this just after the submission deadline).
 We discuss this in more detail in a dedicated reply.

To conclude, we hope that this reply clarifies the main aim of this work (i.e. proposing a novel technique/framework). Once again, we thank the reviewers for their time and efforts.

---

### Meta-Review · Area_Chair_CsRm · 2022-08-22

**Recommendation:** Accept
**Confidence:** Certain

**Metareview:**

PAC-Bayes bounds provides a control of the risk of aggregation of predictors. In these bounds, the Kullback-Leibler divergence between the aggregation distribution and a prior appears in the upper bound on the risk. In this paper, the authors prove variants where the KL is replaced by an IPM (Integral Probability Metrics), including the total variation distance and Wasserstein. The important point is that these bounds are close to "uniform" (Vapnik-type) bounds in some unfavorable settings, but also can improve on them is more favorable scenarios.

The reviewers agreed that the results are novel and that the paper is technically sound. These bounds really extend the framework of PAC-Bayes bounds (for example, they are not necessarily vacuous when the posterior is not absolutely continuous with respect to the prior), and the fact that they recover uniform bounds in the worst case is also nice. All the reviewers recommended to accept the paper, and I agree with them.

The reviewers pointed out a few missing references, I will ask the authors to include them in the paper as promised during the discussion. I will add the following references that were not mentioned by the reviewers, and thus leave the authors decide to include them or not:
- Alquier and Guedj (2018) actually provided PAC-Bayes bounds based on f-divergences, that were then improved by Ohnishi and Honorio (2021) (especially the dependence with respect to the confidence level).
- there were a few attempts to replace the KL by the Wasserstein distance. The benefit were not as clear as in the present paper, but the authors might want to comment on the paper https://hal.archives-ouvertes.fr/hal-03262687/ or Lopez & Jog (2018) on MI bounds...

**Award:**

No

---

### Decision · Program_Chairs · 2022-09-14

Accept